# GM-CSF-dependent CD301b$^+$ mouse lung dendritic cells confer tolerance to inhaled allergens

Christina L. Wilkinson[1], Keiko Nakano [1], Sara A. Grimm [2], Gregory S. Whitehead[1], Yukitomo Arao[3], Perry J. Blackshear [3], Peer W. Karmaus[1], Michael B. Fessler [1], Donald N. Cook [1] ✉ & Hideki Nakano [1] ✉

The severity of allergic asthma is driven by the balance between allergen-specific T regulatory (Treg) and T helper (Th)2 cells. However, it is unclear whether specific subsets of conventional dendritic cells (cDCs) promote the differentiation of Tregs. We have identified a subset of lung resident type 2 cDCs (cDC2s) that display high levels of CD301b and have potent Treg-inducing activity ex vivo. Single-cell RNA sequencing and adoptive transfer experiments show that during allergic sensitization, many CD301b$^+$ cDC2s transition in a stepwise manner to CD200$^+$ cDC2s that selectively promote Th2 differentiation. GM-CSF augments the development and maintenance of CD301b$^+$ cDC2s in vivo, and also selectively expands Treg-inducing CD301b$^+$ cDC2s derived from bone marrow. Upon their adoptive transfer to recipient mice, lung-derived CD301b$^+$ cDC2s confer immunological tolerance to inhaled allergens. Thus, GM-CSF maintains lung homeostasis by increasing numbers of Treg-inducing CD301b$^+$ cDC2s.

Allergic asthma is a widespread disease characterized by reversible airway obstruction, inflammation, and airway hyper-responsiveness (AHR)[1]. Many asthmatics have predominantly eosinophilic inflammation of the airway and have high levels of the type 2 cytokines, IL-4, IL-5, and IL-13, which promote IgE production, eosinophilic inflammation, mucus production, and AHR. Allergen-specific T helper (Th)2 cells are a major source of these cytokines. During allergic sensitization, these cells develop from naïve CD4$^+$ T cells that interact with conventional dendritic cells (cDCs) presenting allergen-derived peptides in the context of MHC class II (MHC-II). However, the presence of allergen-specific Th2 cells does not always lead to allergic disease because peripheral T cells in healthy individuals and in asthmatics recognize the same allergen-specific epitopes[2]. One critical difference between asthmatics and individuals with healthy airways is that the former lack

sufficiently strong regulatory mechanisms to hold allergen-specific effector responses in check[3]. Accordingly, there has been considerable interest in increasing the strength of allergen-specific regulatory responses through subcutaneous immunotherapy (SCIT) or sublingual immunotherapy (SLIT) with the provoking allergens[4]. Although often effective, the allergen dosing must be continuously maintained for 3 years[5], and this has led to poor patient compliance and reduced efficacy of the therapy[6]. Thus, there is an unmet need for immunotherapeutic strategies that have a shorter timeline and consequent improved patient compliance.

A wealth of evidence has shown that regulatory CD4$^+$ T cells (Tregs) can strongly suppress allergic responses[7], including established inflammation of the airway[8]. These cells are characterized by their cell surface display of CD25 (IL-2 receptor alpha subunit) and the

[1]Immunity, Inflammation and Disease Laboratory, Division of Intramural Research, National Institute of Environmental Health Sciences, NIH, Research Triangle Park, NC, USA. [2]Integrative Bioinformatics Support Group, Division of Intramural Research, National Institute of Environmental Health Sciences, NIH, Research Triangle Park, NC, USA. [3]Signal Transduction Laboratory, Division of Intramural Research, National Institute of Environmental Health Sciences, NIH, Research Triangle Park, NC, USA. ✉e-mail: cookd@niehs.nih.gov; nakanoh@niehs.nih.gov

Treg master transcription factor, Foxp3[9]. Several studies have shown that asthma patients have fewer CD25+Foxp3+ Tregs in the peripheral blood than non-asthmatic individuals[10,11]. Although TGF-β and IL-2 have been shown to promote Treg development in vitro[12], much less is known about how these cells arise in vivo. In particular, it is unclear whether a specific subset of lung cDCs promotes allergen-specific Treg development or whether all lung cDCs have this capacity, depending on the environmental signals they receive. The identification of a dedicated Treg-inducing cDC subset should facilitate mechanistic studies of Treg development in vivo and might lead to improved DC-based immunotherapies for asthma.

The lungs of humans and mice contain two major cDC subsets, usually referred to as cDC1 and cDC2[13]. Both of these cDC subsets develop from FMS-like tyrosine kinase 3 ligand (FLT3L)-dependent DC precursors (preDCs)[14] and are therefore developmentally distinct from monocyte-derived macrophages that arise independently of FLT3L[15]. Mouse cDC1s are homogeneous and can be readily identified by their display of CD103 (αE integrin) and relatively low amounts of CD11b (αM integrin). By contrast, mouse cDC2s are heterogeneous, have low amounts of CD103, and display high levels of CD11b. The heterogeneity of cDC2s has confounded efforts to assign specific functions to these cells, but the advent of single-cell RNA-sequencing (scRNA-seq) has provided some much-needed clarity[16,17]. Our group recently reported that the cell surface marker CD301b (encoded by Mgl2) is present on the cell surface of most lung cDC2s at steady state[18]. However, during allergen/adjuvant-mediated allergic sensitization through the airway, additional subsets of cDC2s accumulate in the lung. One such subset displays Ly6C and potently stimulates Th17 differentiation, whereas a different subset has high display levels of CD200 and drives Th2 differentiation[18].

In this work, we studied lung cDCs that promote Treg differentiation and confer allergen-specific tolerance. We show that CD301b+ cDC2s strongly promote the development of allergen-specific CD25+Foxp3+ Tregs in the lung. However, during house dust extract (HDE)-mediated allergic sensitization, many CD301b+ cDC2s transition to more mature Th2-inducing cDC2s. The number of Treg-inducing cDCs is controlled in part by the cytokine granulocyte-macrophage colony-stimulating factor (GM-CSF) because mice lacking Csf2rb in cDCs have fewer CD301b+ cDC2s and more Th2-inducing cDCs. These findings show that GM-CSF maintains immunotolerance in the lung by increasing numbers of Treg-inducing cDC2s.

## Results

### Lung CD301b+ cDC2s induce Treg differentiation ex vivo and immunotolerance in vivo

It has long been known that mice become immunotolerant to the experimental allergen, ovalbumin (OVA), upon inhaling aerosols of that protein[19]. We confirmed this, and further showed that mice receiving oropharyngeal (o.p.) aspirations of highly purified OVA prior to allergic sensitization also become tolerant to OVA (Fig. 1a). Thus, although mice that received o.p. aspirations of HDE mixed with OVA became sensitized to OVA and developed robust allergic airway inflammation upon subsequent OVA challenge, mice that had received o.p. aspirations of purified OVA prior to OVA/HDE sensitization had clearly diminished responses to OVA challenge (Fig. 1b). In particular, eosinophil and neutrophil inflammation were virtually abolished by pretreatment with purified OVA prior to OVA/HDE sensitization. We therefore concluded that this model of allergen-specific tolerance would be appropriate for studying tolerogenic cDCs in the lung.

To determine whether a specific cDC subset preferentially induces Treg differentiation, we treated mice with tolerance-inducing, purified OVA and used antibodies against the subset-specific cell surface markers to purify the various cDC2 subsets by flow cytometry. Total cDCs were identified as CD11c+MHCII+CD88−F4/80−SiglecF−, thus excluding alveolar macrophages (SiglecF+CD88+F4/80+), interstitial macrophages

(CD88+F4/80+), monocytes (F4/80+) and neutrophils (CD88+) (Fig. S1a). Within this total cDC population, cDC1s were identified as CD103+CD11blo and cDC2s as CD103−CD11bhi. cDC2s were further stratified to CD301b+ cDC2s and CD301b− cDC2s. The two cDC2 populations, as well as cDC1s, were cultured ex vivo separately with naïve CD4+ T cells isolated from OT-II OVA-specific T cell receptor transgenic mice bearing a Foxp3eGFP reporter gene to allow Treg detection by flow cytometry (Fig. S1b). After 5 days of coculture, CD103+ cDC1s, as well as CD301b+ and CD301b− cDC2s, had activated OVA-specific CD4+ T cells, as indicated by the increased display of the activation marker CD44. Treg cells, identified by their display of CD25 and GFP fluorescence encoded by the Foxp3eGFP gene, were more efficiently induced by CD301b+ cDC2s than by any of the other cDC subsets (Fig. 1c). These data show that at least under tolerogenic conditions, CD301b+ lung cDC2s strongly promote Treg induction ex vivo.

Although instillation of highly purified OVA efficiently promotes immunotolerance, most allergen preparations are contaminated with adjuvants in the form of bacterial products such as LPS and flagellin[20]. These products, as well as HDE, possess adjuvant activity and can promote allergic sensitization to co-inhaled OVA[20]. However, Treg induction and tolerogenic responses to inhaled allergens can occur even in the presence of adjuvants[21,22]. The profile of cDC2 subsets in the lung changes during allergic sensitization, as Ly6C+ inflammatory cDC2s are recruited to the lung and CD200+ cDC2s undergo expansion[18]. It remained possible, therefore, that the cDC subset(s) that drive regulatory responses during adjuvant-mediated allergic sensitization are different from the CD301b+ cDC2 subset that promotes Treg differentiation at steady state. To test this, we purified cDC subsets from lungs of mice that had been sensitized by o.p. aspirations of OVA/HDE (Fig. S1c), and cocultured these cells with naïve CD4+ T cells from Foxp3eGFP OT-II reporter mice. CD301b+ cDC2s promoted significantly more CD25+ and Foxp3eGFP Tregs compared with all other subsets, including Ly6C+ cDC2s (Fig. 1d). Ly6C+ cDC2s and CD200+ cDC2s are known to predominantly promoted Th17 and Th2 differentiation, respectively[18]. Taken together, these results demonstrate that individual cDC2 subsets in the lung have unique functions and that CD301b+ cDC2s promote Treg differentiation not only at steady state, but also during allergic sensitization.

As Foxp3+ Tregs are composed of multiple subsets[23], we next characterized CD301b+ cDC2-induced Tregs by analysis of their transcription factors. HELIOS is reported to stabilize the suppressive function of Tregs[24], and GATA3 and RORγt are expressed in Th2-type Tregs and Th17-type Tregs, respectively[25]. We found that the majority of Foxp3+ Tregs induced by CD301b+ cDC2s express HELIOS, a smaller fraction expresses GATA3, and relatively few cells express RORγt (Fig. S2a and b). HELIOS was originally found in thymus-derived Tregs (tTregs)[26], but recent evidence has demonstrated the presence of HELIOS in periphery-induced Treg (pTregs), and these cells confer immunotolerance[27]. The co-expression of Foxp3 and HELIOS in Tregs induced by CD301b+ cDC2s suggests that they are likely immunosuppressive.

The unique ability of cDCs to activate naïve T cells has been effectively harnessed in DC-based vaccines. The best-known examples of this are therapies in which tumor antigen-bearing DCs are injected into cancer patients to strengthen their anti-tumor immunity[28]. In allergic diseases such as asthma, the opposite effect is desired, namely, to diminish the strength of immune responses to provoking allergens. To test the feasibility of using DC-based therapy to confer tolerance to allergens in vivo, we purified CD301b+ or CD301b− cDC2s from naïve wildtype (WT) mice, incubated these cells with OVA323-339 peptide and adoptively transferred the cells to naïve mice by o.p. aspiration. One week later, the mice were subjected to the OVA/HDE model of allergic asthma and their cellular inflammation in airways evaluated (Fig. 1e). Mice that had received OVA peptide-loaded CD301b+ cDC2s prior to allergic sensitization had significantly fewer inflammatory cells,

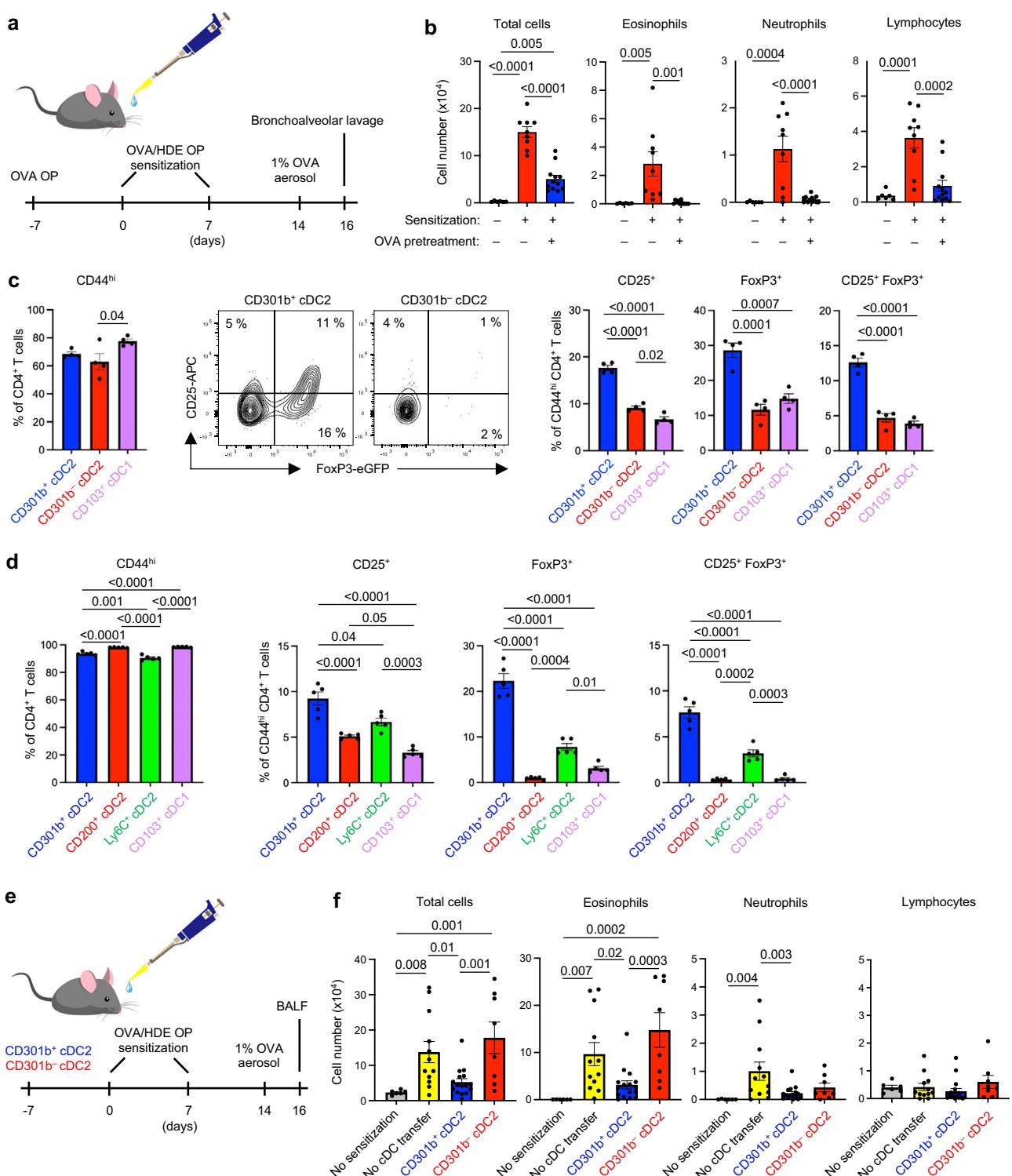

including eosinophils and neutrophils, in the airway after challenge compared with mice that did not receive the cDC pretreatment (Fig. 1f). By contrast, the inflammation in mice that received OVA peptide-loaded CD301b⁻ cDC2s was comparable to that of animals that did not receive cDC pretreatment. Thus, adoptive transfer of allergen-loaded CD301b⁺ cDC2s specifically confers tolerance to inhaled allergens in vivo.

We previously reported that lung CD200⁺ cDC2s from OVA/HDE-sensitized mice promote ex vivo Th2 cytokine production from CD4⁺ T cells[18]. We confirmed this result (Fig. S2c) and further tested whether adoptive transfer of these DCs can exacerbate airway inflammation in a

mouse model of asthma. We found that mice that had received CD200⁺ cDC2s from OVA/HDE-sensitized mice had significantly more total cells, eosinophils, and neutrophils as compared to mice that did not receive cDC pretreatment (Fig. S2d). Together, these results show that CD301b⁺ cDC2s promote tolerance in vivo, whereas CD200⁺ DC2s promote allergic inflammation.

## Lung resident CD301b⁺ cDC2s transition to Th2-inducing cDCs during allergic sensitization

To better understand the relationships between Treg-inducing CD301b⁺ cDC2s and other cDC2 subsets in the lung, we employed

**Fig. 1 | CD301b⁺ lung cDC2s promote Treg differentiation and induce immunological tolerance. a** Timelines for mouse models of asthma. Mice were sensitized with OVA/HDE twice by o.p. aspiration and challenged once with OVA aerosol. BALF was harvested 48 h post-challenge. Tolerance assays were similar, except that mice were also exposed to OVA alone by o.p. aspiration on day -7. **b** Cell numbers for the indicated leukocytes in BALF, as determined by microscopy (No treatment *n* = 6, Sensitization *n* = 9, OVA pretreatment *n* = 12 biological replicates). **c** Activation of CD4⁺ T cells (left), representative cytograms of CD4⁺CD44^hi T cells (middle) and compiled data of Tregs (right) are shown. Naïve CD4⁺ T cells from *Foxp3^eGFP* OT-II mice were cocultured for 5 days with indicated cDC2 subsets isolated from C57BL/6 mice that received OVA by o.p. aspiration (*n* = 4 technical replicates). Gating strategy for purified cDC subsets and CD4⁺ T cell analysis (CD4⁺CD3ε⁺MHCII⁻Live/Dead⁻) is shown in Figure S1a and b. **d** Activation of CD4⁺ T cells (left) and Treg induction in CD44⁺ CD4⁺ T cells (right) by distinct cDC subsets isolated from mice that received OVA/HDE by o.p. aspiration (*n* = 5 technical replicates). Gating strategy for purified cDC subsets is shown in Fig. S1c. **e** Timeline

for mouse model of asthma to test tolerogenic function of cDC2 subsets. CD301b⁺ or CD301b⁻ cDC2s were purified and incubated with OVA₃₂₃₋₃₃₉ peptides, then adoptively transferred by o.p. aspiration to C57BL/6 mice on day 0. After OVA/HDE sensitization and OVA challenge, cells in BALF were analyzed. **f** Cell numbers of the indicated leukocytes in BALF (No sensitization *n* = 6, No cDC transfer *n* = 12, CD301b⁺ cDC2 *n* = 16, CD301b⁻ cDC2 *n* = 8 biological replicates). **b–d** Data were analyzed by one-way ANOVA with Tukey's multiple comparison test. (**f**) Data were analyzed by one-way ANOVA with Fisher's LSD test. **b, f** Each dot represents an individual mouse. Combined results from two independent experiments are shown. (**c, d**) Each dot represents separately cultured CD4⁺ T cells. Representative results from 2 independent experiments are shown. Data are presented as mean values ± SEM. *P* values are indicated above the graphs. Source data are provided as a Source Data file. cDC conventional dendritic cells, Treg regulatory CD4⁺ T cells, OVA ovalbumin, HDE house dust extract, OP oropharyngeal, BALF bronchoalveolar lavage fluid.

scRNA-seq with cellular indexing of transcriptomes and epitopes (CITE-Seq). Mice were harvested at 0 h (steady state), 6 h, and 18 h post-allergic sensitization with OVA/HDE, and total CD11b⁺ lung cDC2s were prepared and analyzed (Figs. 2a, S3a). Cells from the three preparations were pooled and sequenced simultaneously, but different barcodes were used for each time point, allowing us to assign a specific time point to each cell during the analysis. Uniform Manifold Approximation and Projection (UMAP) revealed 12 clusters including 9 cDC2 clusters (Fig. 2b). *Mgl2*-expressing cells and protein CD301b⁺ cells in clusters 8 and 4 were the predominant cDC2s at baseline and were therefore designated 'lung resident' cDC2s (Figs. 2c and d). Clusters 2 and 5 did not become major clusters until 6 h and 18 h post-sensitization, respectively, with cells in both clusters expressing *Cd200*, as well as the *Ccr7* chemokine receptor gene (Figs. 2c and d, S3b). Clusters 1, 3, 6 and 7 also appeared post-sensitization and displayed Ly6C (Figs. 2c and d, S3c). A dendrogram displaying similarity of lung cDC clusters indicate 3 groups; *Mgl2*⁺ (C4 and 8), *Cd200*⁺ (C2 and 5), and Ly6C⁺ (C1, 3, 6 and 7) (Fig. S3d), which are consistent with the UMAP. Transcriptome comparison between mouse lung cDCs in the present study with human DCs from a previous study[29] suggests similarity between human and mouse cDC subsets. The comparison revealed that cluster 7 in Ly6C⁺ mouse cDC2s is similar to monocyte-derived human DC3s, and cluster 4 in *Mgl2*⁺ mouse cDC2s is similar to human DC2 (Fig. S3e).

We next analyzed the scRNA-Seq data by RNA velocity[30], which is used to infer maturation stages of cells based on relative amounts of spliced and unspliced RNA. This approach suggested that during OVA/HDE-mediated allergic sensitization, multiple differentiation events occur simultaneously. Thus, *Mgl2*-expressing lung resident cDC2s in cluster 4 give rise to *Cd200*-expressing cDC2s in cluster 2 (Figs. 2e, S3f). The latter appear to be a transitional cell type linking CD301b⁺ cDC2s with the more mature CD200⁺ cDC2s in cluster 5 (Fig. 2c, d). By contrast, *Mgl2*-expressing lung resident cDC2s in a different cluster (cluster 8) transition to proliferating DCs in cluster 10 (Fig. S3f). In parallel with these changes, newly recruited *Ly6c*-expressing cDC2s in cluster 6 transition to cluster 1 and 2 (Figs. 2e, S3f). The latter observation suggests that during allergic sensitization, *Cd200*⁺ transitional cDC2s in cluster 2 can arise either from CD301b⁺ (*Mgl2*⁺) cDC2s or from newly recruited Ly6C⁺ cDC2s. Overall, allergic sensitization through the airway triggers an increase in Ly6C⁺ inflammatory cDC2s and CD200⁺*Ccr7*⁺ migratory cDC2s, with a corresponding decline in lung resident CD301b⁺ cDC2s (Figs. 2f, g, S4a). However, the Ly6C⁺ cDC2s that arise post-sensitization can also develop into CD301b⁺ cDC2s, suggesting a pathway for replenishing the lung resident cDC2 population and thus maintaining homeostasis.

To experimentally confirm the developmental trajectory suggested by the RNA velocity analysis, we performed DC adoptive transfers with specific cDC2 subsets. CD301b⁺ cDC2s were prepared

from OVA/HDE-sensitized CD45.2 donor mice and transferred into naïve CD45.1 recipient animals (Figs. 2h, S4b). By one day post-transfer, all CD301b⁺ donor DCs had gained CD200 and many had lost CD301b (Fig. 2i). By 3 days post-transfer, all donor cells were negative for CD301b, indicating a complete conversion of donor CD301b⁺ cDC2s to CD200⁺ cDC2s.

We used similar adoptive transfer experiments to study the developmental potential of Ly6C⁺ donor cDC2s. At one day post-transfer, Ly6C⁺ donor cDC2s had reduced display of Ly6C, no increase in CD301b, but marked increased display of CD200 (Fig. S4c, d). These results suggest that the majority of lung cDC2s are in the same lineage, which is in agreement with a recent report demonstrating that the type B cDC2 (cDC2B) lineage is the dominant cDC2 population in the lung and that these cells descend from a common preDC2 progenitor[17].

## Tregs can be induced in the lung

The absence of *Ccr7* expression in CD301b⁺ cDC2s (Fig. S3b) suggested that these cells are non-migratory. To test this, we labeled lung cDCs in situ by instilling the fluorescent dye, PKH26 (PKH), into the airways of mice. On the following day, lung-draining mediastinal lymph nodes (mLNs) were harvested and PKH⁺ migratory cDCs were identified by flow cytometry (Fig. S5a). Although a small number of migratory cDCs were detected in mLNs following treatment of mice with OVA alone, these cells were markedly increased in animals that had been sensitized with OVA/HDE (Fig. 3a). Furthermore, under all conditions tested, the vast majority of migratory cells in mLNs were CD200⁺, with very few CD301b⁺ cDC2s detected (Fig. 3b). These results are consistent with a previous report[18] and confirm that CD301b⁺ cDC2s are lung resident, non-migratory cells.

To test whether DC migration is essential for Treg induction under tolerogenic conditions, we adoptively transferred naïve CD4⁺ T cells from CD45.1 OT-II mice into naïve CD45.2 mice. Foxp3⁺ and CD25⁺ Tregs among activated (CD44^hi) CD45.1⁺ donor CD4⁺ T cells were evaluated by flow cytometry following OVA inhalation without adjuvant in the recipient mice with or without pertussis toxin (PTX) treatment. Although PTX efficiently blocked cDC migration from the lung to draining LNs (Fig. S5b) as previously reported[31], OVA inhalation induced Tregs in PTX-treatment mice at a comparable level with PTX-untreated mice (Fig. 3c). This suggests that cDC2 migration to LNs is dispensable for Treg induction.

To confirm this result, we used *Ccr7*⁻/⁻ mice, whose dendritic cells are unable to migrate from peripheral tissue to LNs. We adoptively transferred naïve CD4⁺ T cells from CCR7-sufficient CD45.1 OT-II mice into WT and *Ccr7*⁻/⁻ mice, and numbers of Foxp3⁺ and CD25⁺ Tregs within the CD45.1 donor cell gate were evaluated following treatment of the recipient animals with OVA alone (Fig. S5c). As expected, development of CD44^hi effector CD4⁺ T cells and Tregs in the lungs of

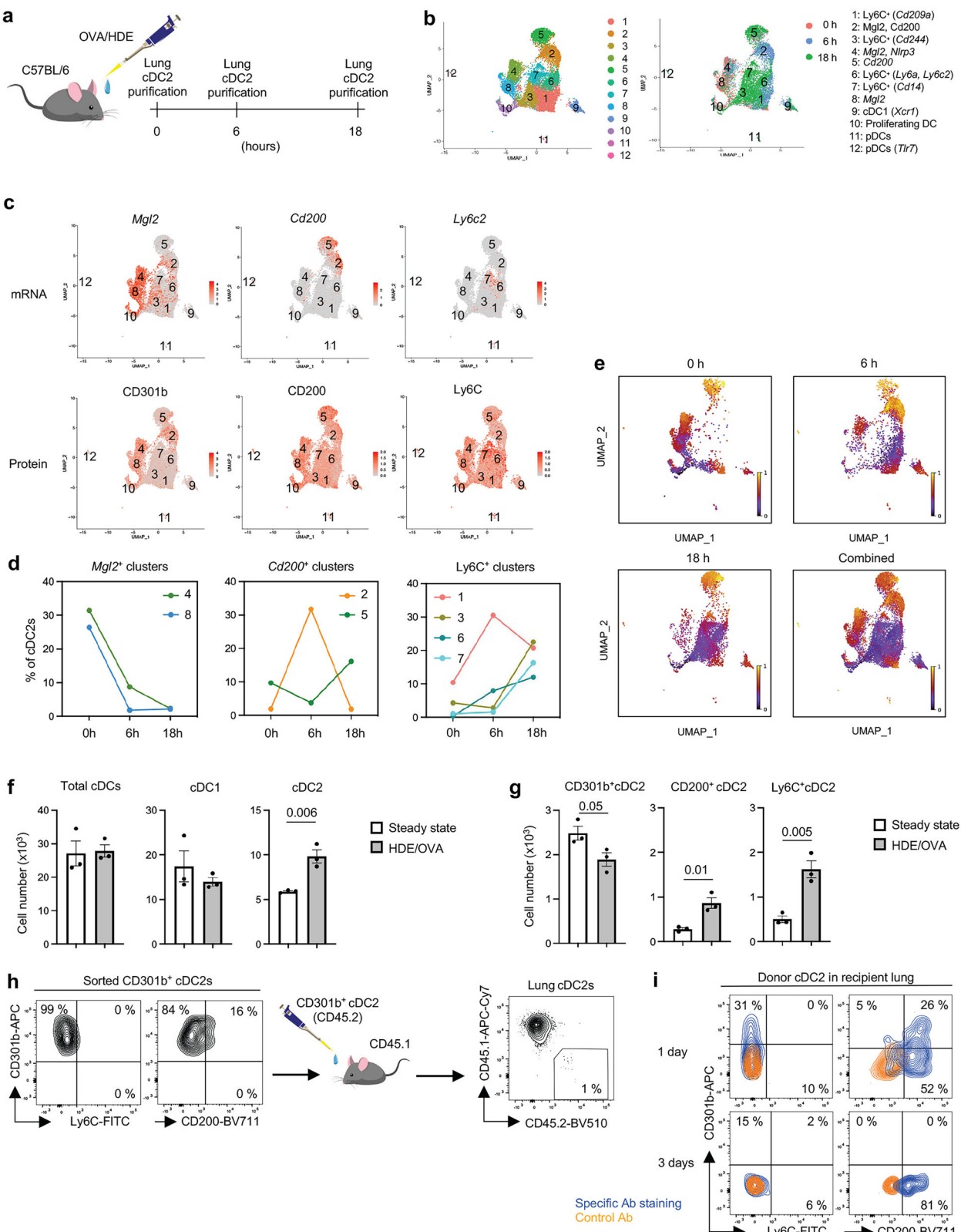

WT mice was dependent on OVA treatment (Fig. 3d). The frequency of OVA-specific CD45.1 Tregs in *Ccr7*[−/−] mouse lungs was as high, or higher, than that seen in WT mice, suggesting that Tregs can be generated in the lung in the absence of cDC migration. These results prompted us to further test whether allergen-specific immunological tolerance can also be induced without cDC migration by exposing *Ccr7*[−/−] mice to OVA prior to allergic sensitization (Fig. 3e).

Pretreatment with OVA by o.p. significantly reduced inflammatory cell accumulation in both WT and *Ccr7*[−/−] compared with mice that were sensitized and challenged without pretreatment (Fig. 3f), although inflammatory cells were slightly more abundant in tolerance-induced *Ccr7*[−/−] mice than in their WT counterparts. Taken together, our results suggest that Tregs can be induced in the lung by non-migratory cDC2s.

**Fig. 2 | CD301b⁺ lung resident cDC2s give rise to CD200⁺ migratory cDC2s.**
**a** Timeline for lung cDC2 isolation following OVA/HDE sensitization. Lung cells were harvested at 0 h (steady state), 6 h, and 18 h post-sensitization, and purified cDC2 were analyzed by CITE-Seq. **b**, UMAP plots showing cDC2 clusters identified in scRNA-Seq (left panel). Clusters are denoted by collection timepoints (right panel). **c** UMAPs showing expression of subset-marker genes (top) and proteins (bottom). **d** Time course of each cluster's abundance in cDC2s at steady state and after allergic sensitization. **e** RNA velocity analysis of cDC2s at the indicated time points, as well as all time points combined. Maturation stages inferred by scVelocity latent time are indicated by colors ranging from unspliced immature RNA (purple) to spliced mature RNA (yellow). **f, g** Abundance of lung cDCs at steady state (white) and 16 h after sensitization (grey) with OVA/HDE were calculated based on total cell counts and flow cytometric analysis. Each dot represents an individual mouse. Data

were analyzed by two-tailed unpaired *t*-test (*n* = 3 biological replicates) and presented as mean values ± SEM. *P* values are indicated above the graphs. **h** Adoptive transfer of purified CD301b⁺ lung cDC2s from OVA/HDE sensitized C57BL/6 mice (CD45.2) to naïve CD45.1 recipient mice. Representative cytograms of purified donor cells and recipient lung cDC2s post-transfer are shown. **i**, Lung cDC2s at days 1 and 3 post-transfer were analyzed by flow cytometry. Cytogram showing the phenotype of donor CD301b⁺ cDC2s-derived cells at the indicated time points. **a, f–h** Gating strategies depicted in Figs. S3a, S4a, b. **f–i** Representative results from 2 independent experiments are shown. Source data are provided as a Source Data file. cDC conventional dendritic cells, OVA ovalbumin, HDE house dust extract, CITE-Seq cellular indexing of transcriptomes and epitopes by sequencing, UMAP uniform manifold approximation and projection, scRNA-Seq single cell RNA sequencing, scVelocity single cell velocity.

## CD301b⁺ cDC2 development is promoted by GM-CSF

The majority of cDCs, including cDC2s, depend on the growth factor FLT3L for their development[14]. In addition, previous studies have revealed that GM-CSF is also required for homeostasis of cDC1s and cDC2s in multiple tissues, including lung[32]. However, it is unknown whether some cDC2 subsets are particularly responsive to GM-CSF and if that cytokine impacts the relative abundance of different cDC2 subsets. RNA-Seq and scRNA-Seq analyses verified that bone marrow preDCs and mature lung cDCs express the *Csf2* receptor genes, *Csf2ra*, *Csf2rb*, and *Csf2rb2* (Fig. S6a and b). Interestingly, flow cytometric analysis revealed that CD301b⁺ cDC2s display significantly higher levels of CSF2Rα protein than did Ly6C⁺ or CD200⁺ cDC2s either at steady state or after allergic sensitization (Fig. 4a).

Given the high display levels of the GM-CSF receptor on CD301b⁺ cDCs, we studied the in vivo effect of GM-CSF on those cells. To test whether overproduction of GM-CSF affects in vivo numbers of CD301b⁺ lung cDC2s, we employed mice with a conditional deletion of the 75-base pair AU-rich element (ARE) in the 3′ region of the *Csf2* gene. This ARE motif normally destabilizes *Csf2* transcripts to prevent overproduction of GM-CSF, but in *Csf2fx-ARE* mice crossed with mice expressing Cre recombinase under control of the *Meox2* gene promoter (*Csf2ΔARE* mice), the ARE motif is deleted during early embryogenesis and *Csf2* transcripts are stabilized with consequent overproduction of GM-CSF[33]. We confirmed that GM-CSF levels in lung homogenates of *Csf2ΔARE* mice were elevated 2-3 times over those of WT mice and *Csf2fx* mice lacking the *Meox2Cre* gene (Fig. 4b). Flow cytometric analyses of lung cDCs (Fig. S7a) showed that *Csf2ΔARE* mice have significantly more total cDC2s compared with control mice, while having significantly fewer cDC1s (Fig. 4c). This increase in cDC2s was primarily due to an increase of CD301b⁺ cDC2s, as Ly6C⁺ cDC2s were moderately decreased, and CD200⁺ cDC2s were unchanged in *Csf2ΔARE* mice at steady state (Fig. 4d and e). The increase of CD301b⁺ cDC2s was also seen in *Csf2ΔARE* mouse lungs after OVA/HDE inhalation (Fig. 4f). The increased numbers of CD301b⁺ cDC2s in *Csf2ΔARE* mice raised the question of whether these mice would also have more Tregs in the lung at steady state. Flow cytometric analysis of lung CD4⁺ T cells revealed that Foxp3⁺ Tregs are indeed more abundant in *Csf2ΔARE* mice compared to *Csf2fx* mice (Fig. 4g). These results support the above findings that CD301b⁺ lung cDC2s promote Treg differentiation.

In agreement with previous reports demonstrating that type 2 alveolar epithelial cells (AT2s) are the major source of GM-CSF[34], we confirmed that *Csf2* is expressed in AT2 cells using a previously published dataset[35] (Fig. S6c). Consistent with these observations, we found that CD301b⁺ cDC2s reside in the interstitium around alveolar ducts that are surrounded by alveoli[36] (Fig. 4h). Together, these results suggest that GM-CSF production by AT2 cells promotes the development or survival of CD301b⁺ cDC2 within the interstitial space near alveoli and alveolar ducts. We also observed CD301b⁺ cells near the airway, but the majority of them were likely interstitial macrophages, as they were negative for CD11c and positive for F4/80 (Fig. S7b). Interestingly, cDC1s, which are also GM-CSF-dependent, located

around airway (Fig. S7c), in agreement with a previous report[37]. Thus, for these macrophages and DC1s, other cell types, including type 2 innate lymphoid cells (ILC2s) might be a source of GM-CSF[38].

Having established that overproduction of GM-CSF is sufficient to increase number of CD301b⁺ cDC2 in the lung, we next tested whether GM-CSF is required for the development of these cells. We employed mice bearing a *Itgax^cre* transgene and a floxed *Csf2rb* gene encoding the GM-CSF receptor β chain, CSF2RB. In these animals, *Csf2rb* is selectively deleted in *Itgax*-expressing CD11c⁺ cells, which include all lung cDCs (*Csf2rbΔDC*). In agreement with previous reports[32], macrophages and cDC1s were significantly reduced in *Csf2rbΔDC* mice compared with either WT C57BL/6 mice or *Csf2rbfx* control mice lacking the *Itgax^cre* transgene (Figs. 5a, and S8a). The frequency of total cDC2s at steady state was only slightly increased in *Csf2rbΔDC* mice and no significant differences were seen for Ly6C⁺ cDC2s (Fig. 5b). In humans, GM-CSF-dependent development of DC3 was reported[39], but we did not detect a clear reduction in Ly6C⁺ cDC2s, probably due to limited similarity between human DC3 and mouse Ly6C⁺ cDC2s (Fig. S3e). However, among Ly6C⁻ cDC2s, lung resident CD301b⁺ cDC2s were reduced in *Csf2rbΔDC* mice compared with their WT counterparts (Fig. 5c). Conversely, the frequency of CD200⁺ cDC2s was significantly higher in *Csf2rbΔDC* mice than in control mice. Similar differences between *Csf2rbfx* and *Csf2rbΔDC* mice were seen after allergic sensitization with OVA/HDE (Fig. 5d). Because macrophages are also dependent on CSF2RB, our experiments thus far did not rule out a role for these cells in the development of CD301b⁺ cDC2s. To test this, we performed adoptive transfer of WT alveolar macrophages into *Csf2rbΔDC* mice. This procedure successfully rescued the number of lung macrophages in *Csf2rbΔDC* mice as previously reported[40], but these animals still had significantly fewer CD301b⁺ cDC2s, both at steady state and after allergic sensitization with OVA/HDE (Fig. S8b and c). These results suggest that GM-CSF is required for the development of CD301b⁺ lung resident cDC2s, not through an indirect effect on macrophages, but through a direct effect on CD301b⁺ cDC2s or their precursors.

## GM-CSF signaling is required for Treg induction and suppression of allergic inflammation

Our finding that CD301b⁺ cDC2s are decreased in *Csf2rbΔDC* mice suggested that total cDC2s from these animals might have a decreased capacity to induce Tregs. To test this, we purified total cDC2s from *Csf2rbfx* and *Csf2rbΔDC* mice and separately cultured them with naïve CD4⁺ T cells prepared from *Foxp3^eGFP* OT-II reporter mice. cDC2s from *Csf2rbfx* control mice readily promoted Treg differentiation, as measured by Foxp3^eGFP fluorescence and surface display of CD25, whereas cDC2s from *Csf2rbΔDC* mice were significantly less potent in this regard (Fig. 5e). Conversely, T cells cultured with cDC2s from *Csf2rbΔDC* mice produced more IL-4, IL-5 and IL-13 than did T cells cocultured with cDC2s from control mice, although the increase in IL-5 did not reach statistical significance (Fig. 5f). Adoptive transfer of WT macrophages into *Csf2rbΔDC* mice did not restore WT levels of Treg generation (Figure S8d), suggesting that reduction of Treg induction by *Csf2rbΔDC*

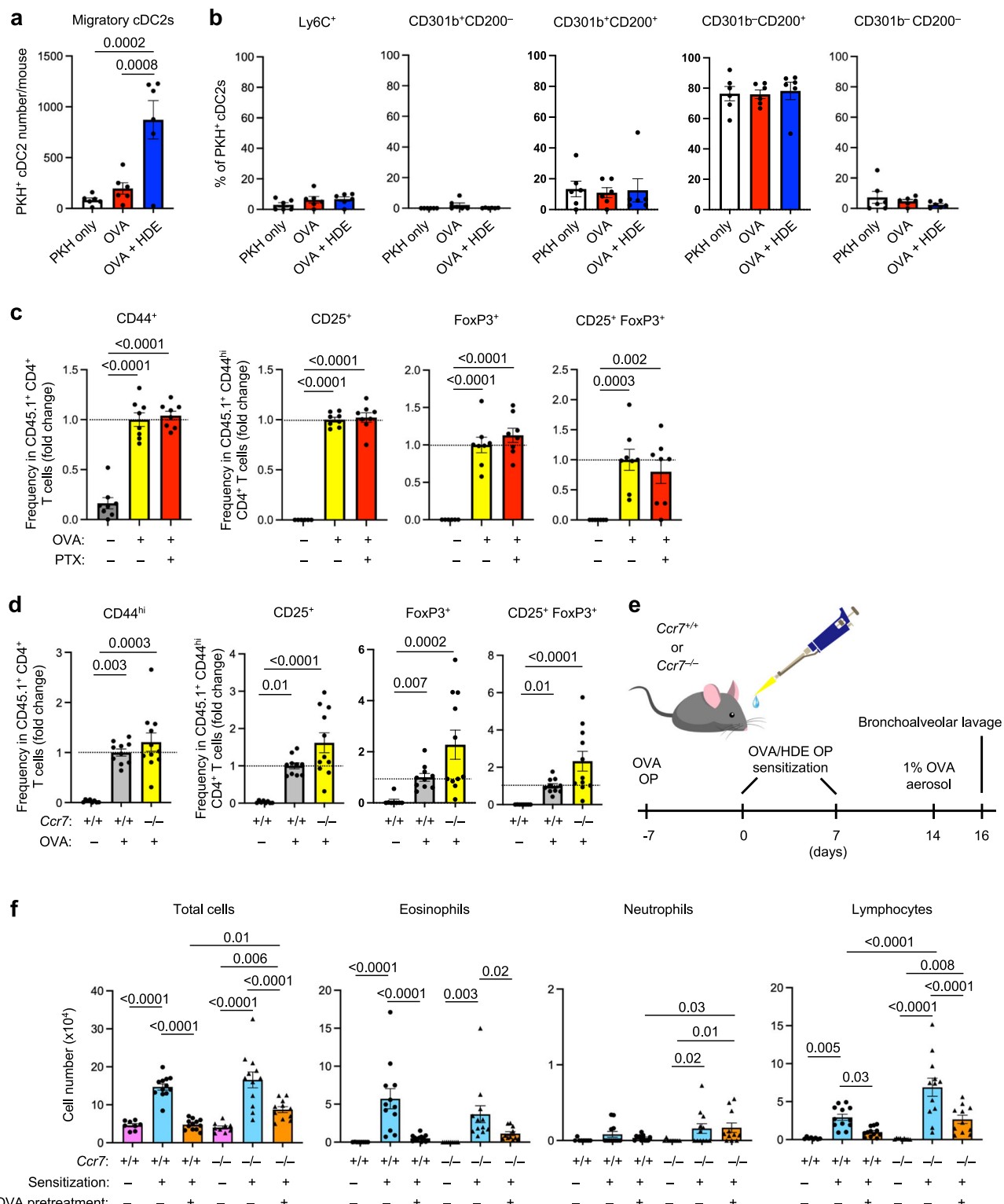

mouse cDC2s is not due to reduced number of macrophages in the mutant mice. Taken together, these data suggest that GM-CSF-dependent cDC2s promote Treg differentiation and suppress Th2 differentiation.

### The role of TGF-β and moderate costimulatory signals in Treg induction by CD301b⁺ cDC2s

Previous study suggests that high levels of costimulatory signals induce Th2 differentiation in vitro, while low level costimulatory signals are associated with Treg induction in skin-draining LNs[41]. In agreement with the report, we found that genes encoding the costimulatory molecules CD40, CD80, CD86, as well as ICAM-1, were expressed at lower levels in *Mgl2*⁺ clusters (C4 and C8) compared with *Cd200*⁺ cDC2 clusters (C2 and C5) (Fig. S3b). These results prompted us to test display levels of costimulatory molecule proteins on the surface of different cDC2s (Fig. S9a). Compared with CD200⁺ cDC2s at steady state, CD301b⁺ cDC2s displayed significantly lower levels of MHC-II, CD40, CD80 and CD86 (Fig. 6a). Similarly, after OVA/HDE inhalation, MHCII, CD40, and CD86 levels were again lower on CD301b⁺ cDC2s than on CD200⁺ cDC2s, while CD80 levels were

**Fig. 3 | Tregs can be induced in the lung without cDC migration. a, b** Migration of cDC2 subsets. All C57BL/6 mice received PKH26 dye and some mice received OVA or OVA/HDE by o.p. aspiration. The frequencies of each subset in PKH$^+$ migratory cDC2s in mLNs of the recipient mice were evaluated by flow cytometry ($n = 6$ biological replicates). Gating strategy depicted in Fig. S5a. **c** Treg generation in the lung of WT mice treated with PTX. All mice received naïve CD4$^+$ T cells isolated from CD45.1 OT-II mice by intravenous injection. OVA and PTX were given by o.p. aspiration to the indicated groups. The phenotype of CD45.1$^+$ donor-derived CD4$^+$ T cells was analyzed by flow cytometry of surface proteins and intracellular Foxp3 (Control: CD44$^{hi}$ $n = 8$, CD25$^+$ $n = 6$, Foxp3$^+$ $n = 6$ CD25$^+$Foxp3$^+$ $n = 7$; OVA $n = 8$; PTX + OVA $n = 8$ biological replicates). **d** Treg generation in the lung of WT and $Ccr7^{-/-}$ mice. All mice received naïve CD4$^+$ T cells isolated from CD45.1 OT-II mice by intravenous injection, and OVA by o.p. aspiration. The phenotype of CD45.1$^+$ donor-derived CD4$^+$ T cells was analyzed by flow cytometry of surface proteins and intracellular Foxp3 (No OVA $n = 8$, $Ccr7^{+/+}$ OVA $n = 10$, $Ccr7^{-/-}$ $n = 11$ biological replicates). **e** Timeline for mouse model of asthma to test tolerance

induction. Some WT and $Ccr7^{-/-}$ mice received OVA by o.p. aspiration and were sensitized with OVA/HDE. After OVA aerosol challenge, cells in BALF were analyzed. **f** Cell numbers of the indicated leukocytes in BALF ($Ccr7^{+/+}$ no treatment $n = 8$, $Ccr7^{+/+}$ sensitization $n = 12$, $Ccr7^{+/+}$ OVA pretreatment $n = 12$, $Ccr7^{-/-}$ no treatment $n = 9$, $Ccr7^{-/-}$ sensitization $n = 12$, $Ccr7^{-/-}$ OVA pretreatment $n = 11$ biological replicates). **a–d, f** Each dot represents an individual mouse. Combined results from two (**a–c, f**) or three (**d**) independent experiments are shown. Data are presented as mean values ± SEM. (**c, d**) Data is presented in the fold change of each subset frequency within the CD45.1$^+$CD4$^+$ or within CD45.1$^+$CD44$^{hi}$CD4$^+$ population. Gating strategy depicted in Fig. S5c. **a–c, f** Data were analyzed by one-way ANOVA with Fisher's LSD test. **d** Data were analyzed by Kruskal-Wallis test with Dunn's multiple comparisons. $P$ values are indicated above the graphs. Source data are provided as a Source Data file. Treg regulatory CD4$^+$ T cells, cDC conventional dendritic cells, OVA ovalbumin, HDE house dust extract, OP oropharyngeal, WT wildtype, PTX pertussis toxin, BALF bronchoalveolar lavage fluid.

---

comparable. These results suggest that low levels of costimulatory molecules might contribute to the Treg-inducing ability of CD301b$^+$ cDC2s. However, Ly6C$^+$ cDC2s also had relatively low display levels of costimulatory proteins, despite the limited ability of these cDCs to promote Treg differentiation (Fig. 1d). This suggested that CD301b$^+$ cDC2s likely have additional features that allow them to promote Treg induction and tolerance.

Immunotolerance is associated with several soluble molecules, including IL-10, IL-35, retinoic acid, and TGF-β[42]. We therefore investigated the potential contribution of those molecules to Treg induction by CD301b$^+$ cDC2s. IL-10 is a well-known immunosuppressive cytokine that decreases secretion of pro-inflammatory mediators from allergen-specific effector T cells[43,44]. However, our scRNA-Seq analysis did not reveal selective expression of *Il10* or the IL-10 receptor genes, *Il10ra* and *Il10rb*, in CD301b$^+$ cDC2s (Fig. S9b). Another cytokine, IL-35, can induce IL-35-producing, inducible Tregs (iTr35)[45]. *Ebi3* encodes the EBI3 subunit of IL-35, and we found that this gene is expressed by the *Mgl2*$^+$ and *Cd200*$^+$ cDC2 clusters C2, C4 and C5, with C2 containing the highest expressers (Fig. S9b). Another molecule, aldehyde dehydrogenase 1A2 (ALDH1A2), can also promote Treg differentiation by generating retinoic acid[46], and *Aldh1a2* is expressed by two *Cd200*$^+$ cDC2 clusters (C2 and C5). The former cluster C5 is CD301b$^-$CD200$^+$, while the later cluster C2 is CD301b$^+$CD200$^+$ (Fig. 2c). Finally, *Tgfb1*, the gene encoding TGF-β, an established inducer of Treg differentiation, is expressed in two *Mgl2*$^+$ cDC2 clusters (C4 and C8). Of note, one of these clusters (C4) also expresses genes encoding the TGF-β-activating factors, FURIN (*Furin*) and leucine-rich repeat containing 33 (LRRC33) (*Nrros*)[47] (Figs. 6b, S9c), suggesting that CD301b$^+$ cDC2s can produce active forms of TGF-β.

Because CD301b$^+$ cDC2 clusters express *Aldh1a2*, *Ebi3*, and *Tgfb1*, we used selective inhibitors to test the requirement of proteins encoded by those genes in Treg induction. When added to cocultures of cDC2s and naïve CD4$^+$ T cells, neither aldehyde dehydrogenase inhibitor nor anti-EBI3 neutralizing antibodies affected Treg generation (Fig. S10a–c). By contrast, Foxp3$^+$ Treg generation ex vivo was suppressed by multiple inhibitors of TGF-β receptor signaling, including SB431542, SD208 and RepSOX (Figs. 6c, d, S10d). These results suggest that TGF-β produced by CD301b$^+$ cDC2s promotes Foxp3$^+$ Treg development.

### Bone marrow-derived CD301b$^+$ cDC2s induce Tregs

Our findings that CD301b$^+$ cDC2s from the lung can induce Treg differentiation ex vivo and confer immunotolerance to inhaled allergens in vivo suggested that those or similar DCs might have potential as immunotherapeutic agents to suppress allergic disease. However, the lung is not a practical source of immunotherapeutic DCs for humans. We therefore conducted a series of experiments to determine whether tolerogenic DC2s can be generated from bone marrow. Given that GM-

CSF can expand numbers of CD301b$^+$ cDC2s in the lung, we tested whether this cytokine can also expand their counterparts in cultures of bone marrow DCs (BMDC2s). We therefore cultured bone marrow cells for 6 days in the presence of FLT3L to differentiate BMDCs, then added GM-CSF (Figs. 7a and S11a). In cultures containing only FLT3L, very few immature CD172a$^+$CD24$^-$ BMDC2s displayed CD301b. However, addition of GM-CSF at day 6 dramatically increased the numbers of those cells at day 7 and 8 (Fig. 7a). By contrast, numbers of CD200$^+$ BMDC2s were minimally affected by the presence of GM-CSF. These data show that, as with CD301b$^+$ lung cDC2s, GM-CSF also selectively increases numbers of CD301b$^+$ BMDC2s.

We next questioned whether GM-CSF is necessary for the development of CD301b$^+$ BMDC2s. After addition of GM-CSF to bone marrow cell cultures, number of CD301b$^+$ BMDC development were significantly reduced in cultures of *Csf2rb*$^{ΔDC}$ mice compared with their counterparts from GM-CSF receptor-sufficient *Csf2rb*$^{fx}$ mice (Fig. S11b). Some CD301b$^+$ DC2s did develop in the *Csf2rb*$^{ΔDC}$ cultures, however, possibly due to the presence of the low affinity GM-CSF receptor beta chain CSF2RB2. Unlike the reduction in CD301b$^+$ BMDC2s in *Csf2rb*$^{ΔDC}$ cultures, CD200$^+$ cells were similar in the two strains. These results suggest that GM-CSF is selectively required for the induction of CD301b$^+$ DC2 development from bone marrow.

To test the helper T cell-inducing activity of BMDC2s, CD301b$^+$ and CD200$^+$ BMDC2 subsets were purified by flow cytometry (Fig. S11c), loaded with OVA$_{323-339}$ peptide, and cocultured with naïve CD4$^+$ T cells from OT-II mice. Purified CD200$^+$ BMDC2s induced more type 2 cytokine production than did CD301b$^+$ BMDCs (Fig. 7b). However, when these same cells were cocultured with CD4$^+$ T cells from *Foxp3*$^{eGFP}$ OT-II reporter mice, both BMDC2 subsets activated T cells, as indicated by their increased display of CD44 and CD25. In agreement with our findings from CD301b$^+$ lung cDC2s, CD301b$^+$ BMDC2s were much more effective at promoting Treg differentiation than were CD200$^+$ BMDC2s, as determined by *Foxp3*$^{eGFP}$ fluorescence in CD4$^+$ T cells (Figs. 7c, S11d). Thus, at least in vitro, the highly accessible CD301b$^+$ BMDC2s function similarly to their lung counterparts and therefore might have great potential for DC-based immunotherapies to treat inflammatory diseases such as asthma.

To test the potential feasibility of using CD301b$^+$ BMDC2s as a therapeutic, we purified these cells, loaded them with OVA$_{323-339}$ peptides, and adoptively transferred them into mice that had been previously sensitized to OVA and HDE (Fig. 7d). One week following the transfer, we challenged the mice with OVA aerosol and measured the cellular accumulation in bronchoalveolar lavage fluid (BALF). We found that CD301b$^+$ BMDC2-treated mice had significantly less eosinophilia compared with mice that did not receive BMDCs, although lymphocytes were moderately increased in CD301b$^+$ BMDC2-treated mice (Fig. 7e). These results provide evidence that CD301b$^+$ BMDCs

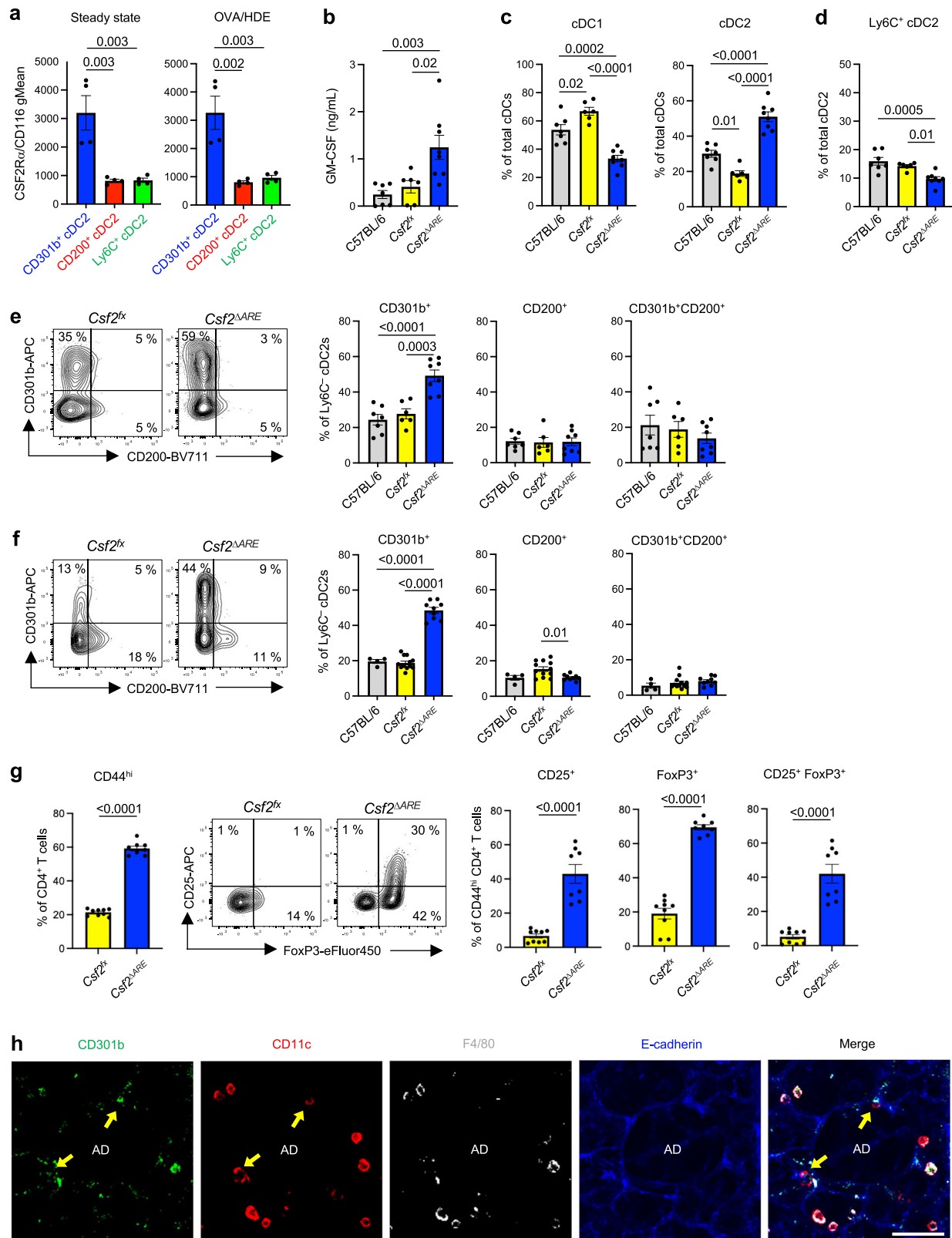

have potential to confer immunotolerance even after allergic sensitization.

## Discussion

Recent research has improved our understanding of the cellular and molecular basis of asthma, but this progress has not led to a corresponding advance in the development of novel and effective therapies.

Thus, inhaled corticosteroids remain the primary treatment for asthmatics. Although these powerful anti-inflammatory drugs are effective in the short term, they affect multiple immune cell types and are therefore not appropriate for long term use. Immunotherapies in the form of SCIT and SLIT are effective; scRNA-seq of peripheral blood mononuclear cells from patients before and after SLIT treatment leads to clonal expansion of allergen-specific Tregs as well as trans-type Th2

**Fig. 4 | GM-CSF promotes the development of lung CD301b⁺ cDC2s. a** Surface display level of CSF2Rα on lung cDC2 subsets at steady state and 16 h after OVA/HDE inhalation were analyzed by flow cytometry (*n* = 4 biological replicates). **b** GM-CSF level in lung homogenates was measured by ELISA (C57BL/6 *n* = 7, *Csf2^fx^ n* = 6, *Csf2^ΔARE^ n* = 8 biological replicates). **c–f** The frequencies of cDC1s and cDC2 subsets in mouse lungs were analyzed by flow cytometry at steady state (**c–e**, C57BL/6 *n* = 7, *Csf2^fx^ n* = 6, *Csf2^ΔARE^ n* = 8 biological replicates) or after OVA/HDE inhalation (**f**, C57BL/6 *n* = 4, *Csf2^fx^ n* = 12, *Csf2^ΔARE^ n* = 9 biological replicates). (**e, f**) Representative cytograms (left panels) and compiled data (right panels) of Ly6C⁻ cDC2s are shown. Gating strategy depicted in Fig. S7a. **g** Lung cells were harvested and stained from mice at steady state. Activated CD4⁺ T cells (left), representative cytograms of CD4⁺CD44^hi^ T cells (middle) and compiled data of Tregs (right) (*Csf2^fx^ n* = 9, *Csf2^ΔARE^ n* = 8 biological replicates). **h** Representative image of an alveolar duct in a mouse lung analyzed by a laser-scanning microscope. Individual signals for CD301b (green), CD11c (red), F4/80 (grey), and E-cadherin (blue) are shown, as well as a merged image. Arrows indicate CD11c⁺CD301b⁺F4/80⁻ cells. Scale bar represents 50 μm. **a–g** Data are presented as mean values ± SEM. Each dot represents an individual mouse. Representative data from two experiments (**h**) or combined data from two experiments (**a–g**) are shown. **a–f** Data were analyzed by one-way ANOVA with Tukey's multiple comparison test. **g** Data were analyzed by two-tailed unpaired *t*-test. *P* values are indicated above the graphs. Source data are provided as a Source Data file. GM-CSF granulocyte macrophage colony stimulating factor cDC conventional dendritic cells, OVA ovalbumin, HDE house dust extract, ELISA enzyme-linked immunosorbent assay, Treg regulatory CD4⁺ T cells, AD alveolar duct.

cells that express high *musculin*, *TGF-β*, and *IL-2*[48]. However, these treatments require several years of continued compliance, and this has hindered their ultimate effectiveness. Thus, the development of novel treatments remains critical for improving patient management and for reversing the symptoms of asthma.

One approach to mitigate the symptoms of asthma is to improve the efficiency of immunotherapy so that shorter regimens can be used. It is well established that Tregs suppress allergic responses and their differentiation is promoted in vitro by the cytokines TGF-β and IL-2[12], however, how these cells develop in vivo is less clear. Our group recently showed that distinct subsets of cDC2s, namely Ly6C⁺ cDC2s and CD200⁺ cDC2s, preferentially promote the differentiation of Th17 and Th2 cells, respectively[18]. Our current findings reveal that yet another cDC2 subset, which displays the surface protein CD301b, is the dominant cDC2 subset in the lung at steady state conditions, and that this subset preferentially promotes Treg differentiation. This result was unexpected because CD301b⁺ cDC2s in skin-draining LNs can stimulate Th2 differentiation[49]. Since lung resident cDC2s downregulate CD301b and upregulate CD200 on their surface upon activation, surface markers of mature cDC2s are likely different between skin and lung.

A role for the IL-10 receptor on cDCs for tolerance induction was previously reported[44], but our scRNA-Seq analysis did not show strong or selective expression of IL-10 receptor in any of the lung cDC2 clusters. Nonetheless, we cannot exclude the possibility of IL-10 acting on CD301b⁺ cDC2s. We did find that CD301b⁺ cDC2s have lower levels of costimulatory molecules, especially CD86, when compared with CD200⁺ cDC2s. This is in agreement with a recent report that strong costimulatory signals from cDC2s promote Th2 differentiation, whereas moderate costimulatory signals promote Treg differentiation[41]. Moreover, *Mgl2*⁺ cDC2 clusters, which contain CD301b⁺ cDC2s, express higher levels *Tgfb1* and genes encoding TGF-β-activating factors than do *Cd200*⁺ cDC2s. These genes include *Furin*, which encodes an endoprotease that cleaves newly synthesized, full-length, TGF-β into two peptides: the latency-associated peptide (LAP) and the mature TGF-β cytokine[50]. CD301b⁺ cDC2s also express *Nrros*, which encodes LRRC33, a protein that tethers TGF-β to the cell surface, where it can be activated by αvβ6 and αvβ8 integrins on neighboring cells. Our finding of *Nrros* expression in CD301b⁺ cDC2s is in agreement with its previously reported expression on antigen-presenting cells[51]. We also found that inhibitors of TGF-β suppressed Treg induction by cDC2s, providing functional evidence of this pathway. Taken together, our data suggest that weak costimulatory signals and production of mature TGF-β by CD301b⁺ lung cDC2s cooperatively induce Treg differentiation.

Our scRNA-seq data and cDC migration assays revealed that tolerance-inducing CD301b⁺ cDC2s are non-migratory, whereas previous reports have shown that DC migration is required for tolerance induction because mice lacking CCR7 developed airway inflammation even after tolerance induction by OVA inhalation[52]. However, in that study, mice were sensitized by intraperitoneal (i.p.) injections of OVA/ alum prior to challenge with OVA aerosol. It is possible that the requirement for cDC migration in tolerance induction is asthma model-dependent or tissue-specific. Also, *Ccr7⁻/⁻* mice and *plt/plt* mice, which lack CCR7 ligands CCL19 and CCL21, are known to develop enhanced inflammation[53,54]. It is possible, therefore, that the enhanced inflammation previously seen in tolerized *Ccr7⁻/⁻* mice is due to their enhanced effector response, rather than impaired immunological tolerance. Nonetheless, our analyses of Tregs following DC migration blockade by PTX and our studies of *Ccr7⁻/⁻* mice both show that cDC migration is not essential for Treg development, and that Tregs can be induced in the lung.

GM-CSF drives proliferation of multipotent myeloid progenitors, resulting in the expansion of myeloid cells, including granulocytes, macrophages and cDCs[55,56]. Previous studies employing GM-CSF receptor-deficient mice revealed the requirement of GM-CSF signaling for cDC homeostasis in several tissues, including the lung. GM-CSF upregulates IRF4[57], a transcription factor required for cDC2 differentiation[58]. However, the requirement for GM-CSF is not absolute, as some cDC2s are present in the lungs of *Csf2r*-deficient mice[32]. This raises the question of whether some cDC2 subsets are particularly responsive to GM-CSF. In the present study, we found that numbers of lung resident CD301b⁺ cDC2s were significantly reduced in *Csf2rb^ΔDC^* mice, whereas migratory CD200⁺ cDC2s were relatively increased. Conversely, overexpression of GM-CSF in *Csf2^ΔARE^* mice led to elevated numbers of lung CD301b⁺ cDC2s compared to WT control animals and no change in CD200⁺ cDC2s. These in vivo results, together with our finding that GM-CSF expands CD301b⁺ cDC2s derived from BM, demonstrate that GM-CSF promotes the expansion of CD301b⁺ cDC2s. Some CD301b⁺ cDC2s remain in lungs of *Csf2rb^ΔDC^* mice, however, suggesting that other growth factors might partly compensate for the absence of GM-CSF. One candidate is M-CSF, as it can partly compensate for FLT3L in *Flt3*-deficient mice[59].

The reduction of CD301b⁺ cDC2s in *Csf2rb^ΔDC^* mice was accompanied by a decrease in the ability of their total cDC2s to stimulate Treg differentiation. Paradoxically, however, GM-CSF is reported to enhance immune responses by augmenting the ability of DCs to take up antigen and prime T helper cells. This cytokine also possesses adjuvant activity and can promote allergic airway inflammation and airway hyperresponsiveness[60,61]. Thus, GM-CSF might enhance the immunostimulatory actions of activated migratory cDCs (including CD200⁺ cDC2s) while simultaneously promoting immune homeostasis by maintaining the pool of lung resident, Treg-inducing cDCs.

The present study reveals that lung resident cDC2s displaying CD301b are the predominant cDC2 subset at steady state, and that these cDCs preferentially induce Treg differentiation from naïve CD4⁺ T cells. Further, we found that the numbers of these cDC2s are maintained by GM-CSF, and that perturbation of this pathway, through over- or underproduction of that cytokine, changes the balance of Tregs and Th2 cells in a model of allergic asthma. These findings improve our understanding of how lung cDC and Treg development are coordinated in vivo. Regarding implications for immunotherapy,

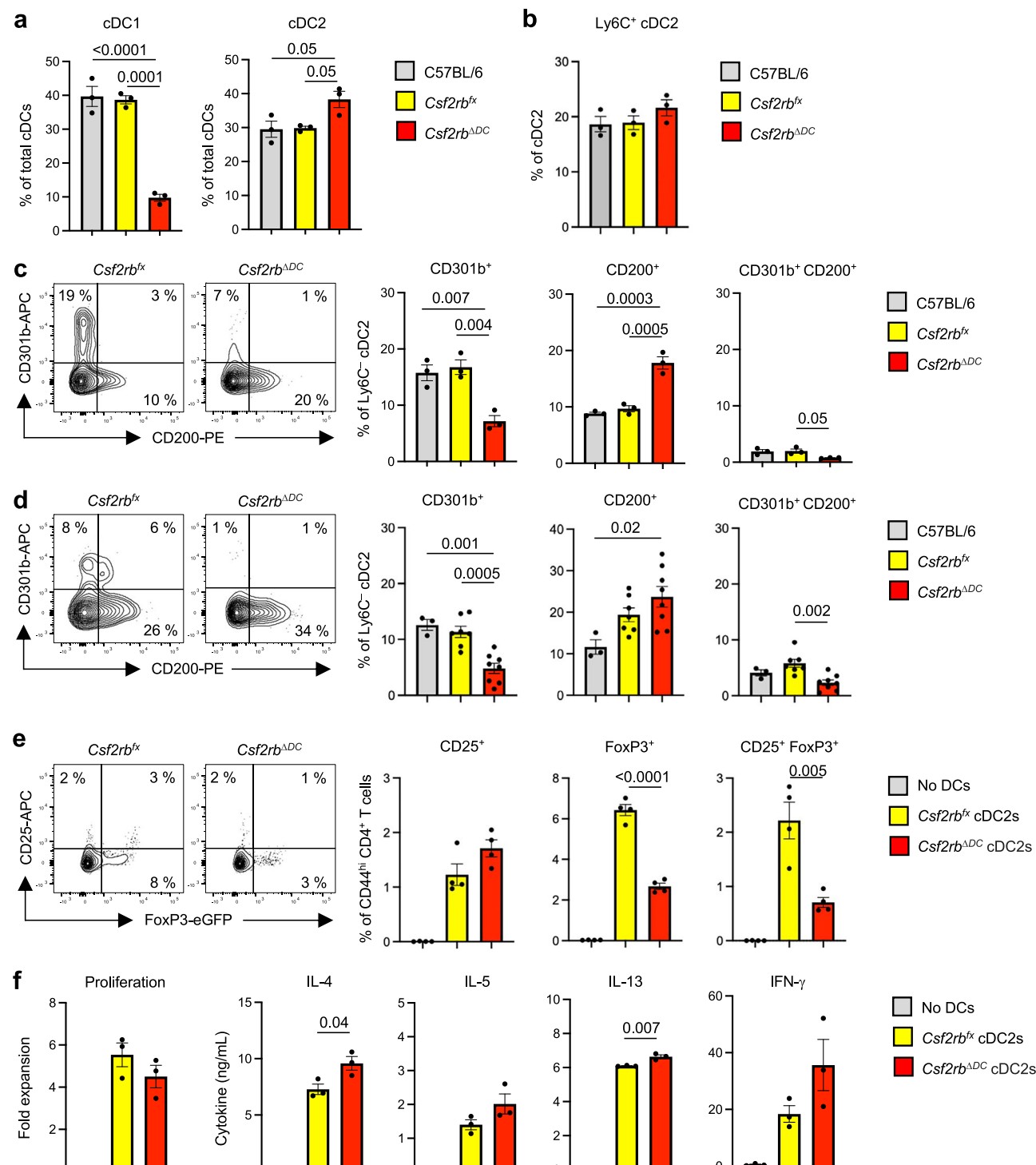

**Fig. 5 | GM-CSF signal is required for CD301b⁺ cDC2 development.** Frequencies of cDC1s and cDC2 subsets in mouse lungs at steady state (**a**−**c**) and after OVA/HDE inhalation (**d**) Representative cytograms (left panels) and compiled data (right panels) of Ly6C⁻ cDC2s are shown (**a**−**c**), *n* = 3 biological replicates; **d**, C57BL/6 *n* = 3, *Csf2rb^fx^* *n* = 7, *Csf2rb^ΔDC^* *n* = 8 biological replicates). **e** Treg induction by lung cDC2s. Flow cytometric analysis of CD25⁺ and Foxp3⁺ Tregs after culture of naïve CD4⁺ T cells from *Foxp3^eGFP^* OT-II mice with or without total lung cDC2s from mice that received OVA (*n* = 4 technical replicates). **f** Effector T cell responses induced by lung cDC2s. Proliferation and cytokine production by T cells, as determined by cell count and ELISA, respectively, after the culture of naïve CD4⁺ T cells from OT-II mice with or without total lung cDC2s mice that received OVA/HDE (*n* = 3 technical

replicates). **a**−**d** Data were analyzed by ordinary one-way ANOVA with Tukey's multiple comparison test. **e**, **f** Data were analyzed by two-tailed unpaired *t*-test. (**a**−**d**) Each dot represents an individual mouse. **e**, **f** Each dot represents separately cultured CD4⁺ T cells. **a**−**c**, e, f Representative results from two experiments (**d**) combined results from two experiments. Comparison between *Csf2rb^fx^* and *Csf2rb^ΔDC^* cDC2s are shown. Data are presented as mean values ± SEM. *P* values are indicated above the graphs. Source data are provided as a Source Data file. GM-CSF granulocyte macrophage colony stimulating factor, cDC, conventional dendritic cells, OVA ovalbumin, HDE house dust extract, Treg regulatory CD4⁺ T cells, ELISA enzyme-linked immunosorbent assay.

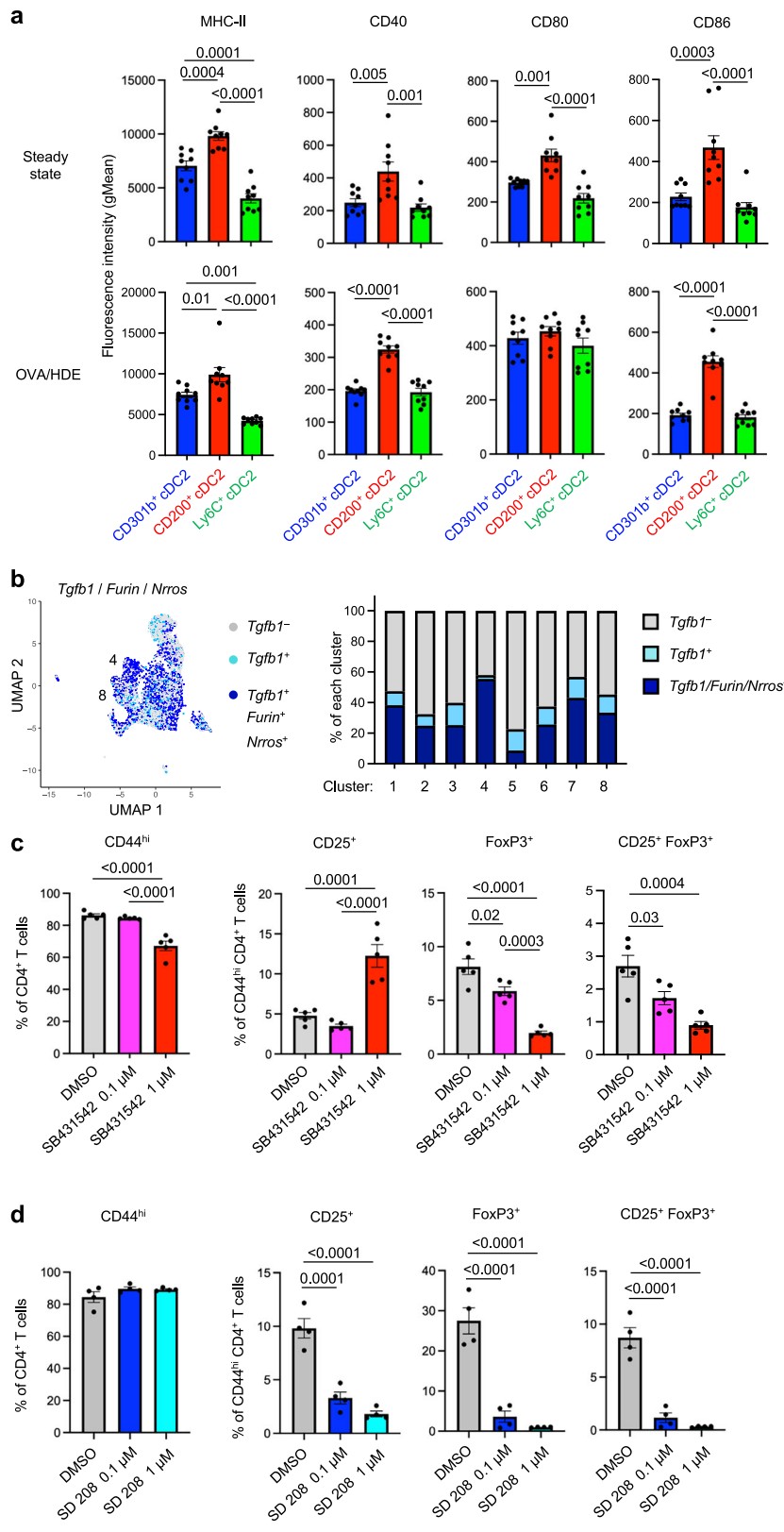

our finding that allergen-bearing CD301b⁺ cDC2s and BMDCs confer immunotolerance when adoptively transferred to naïve or allergically sensitized, respectively, mice shows that this specific DC2 subset might have advantages over other DCs for generating tolerance to inhaled allergen. Further study of these DCs might also reveal other novel pathways that can be therapeutically targeted to treat or even prevent allergic diseases.

## Methods

### Mice

All animal procedures complied with institutional guidelines and were approved by the NIEHS Animal Care and Use Committee (Animal Study Protocols 05-22 and 05-23). C57BL/6 J (stock 000664), *Cd11c^Cre* (B6.Cg-Tg*(Itgax-cre)1-1Reiz/*]); stock 008068), CD45.1 (B6.SJL-*Ptprc^a Pepc^b*/BoyJ; stock 002014), C57BL/6-OT-II TCR transgenic (B6.Cg(TcraTcrb)

**Fig. 6 | Costimulatory molecule levels on cDC2 subsets and the role of TGF-β on Treg induction. a** Surface display levels of MHC-II (I-A) and costimulatory molecules on cDC2 subsets at steady state (top) and 16 h after OVA/HDE inhalation (bottom) were analyzed by flow cytometry (*n* = 9 biological replicates). Gating strategy depicted in Fig. S9a. **b** UMAP of lung cDC2 scRNA-Seq analysis depicting cells expressing *Tgfb1*, *Furin* and *Nrros* (left), and percentages of cells expressing *Tgfb1*, *Furin* and *Nrros* in each cluster (right). Effect of the TGF-β receptor inhibitor, SB43142 (**c**, *n* = 5 technical replicates), or SD 208 (**d**, *n* = 4 technical replicates), on Treg induction by lung cDC2s. CD25 and Foxp3GFP in CD4+ T cells were analyzed by flow cytometry 5 days after coculture with total lung cDC2s. Gating strategy is depicted in Fig. S10a. **a** Each dot represents an individual mouse. Combined results from two independent experiments are shown. **c, d** Each dot represents a separate culture of CD4+ T cells. Representative results from two experiments are shown. **a, c, d** Data were analyzed by one-way ANOVA with Tukey's multiple comparison test. Data are presented as mean values ± SEM. *P* values are indicated above the graphs. Source data are provided as a Source Data file. cDC conventional dendritic cells, TGF-β transforming growth factor beta, Treg regulatory CD4+ T cells, MHC-II major histocompatibility complex class II, OVA ovalbumin, HDE house dust extract, UMAP uniform manifold approximation and projection, scRNA-Seq single cell RNA sequencing.

425Cbn/J; stock 004194), *Foxp3eGFP* (B6.Cg-*Foxp3tm2Tch*/J; stock 006772), *Ccr7−/−* (C57BL/6-*Ccr7tm1.1Dnc*/J; stock 027913), and *Meox2Cre* (B6.129S4-*Meox2tm1(Cre)Sor*/J; stock 003755) mice were purchased from Jackson Laboratories. CD45.1 OT-II mice were bred by crossing CD45.1 and OT-II mice. *Foxp3eGFP* OT-II mice were bred by crossing *Foxp3eGFP* and OT-II mice. *Csf2fx-ARE* mice were generated by Perry Blackshear (NIEHS) as described previously[33]. The mice lacking 75 bp AU-rich element (ARE) in *Csf2* mRNA (*Csf2ΔARE*) were generated by crossing *Meox2Cre* and *Csf2fx-ARE* mice[33]. *Csf2rbfx* (C57BL/6-*Csf2rbtm1c(EUCOMM)Hmgu*/Orl) mice originally generated by Burkhard Becher (University of Zürich, Switzerland) were obtained from the European Mouse Mutant Archive[62]. *Cd11cCre/wt Csf2rbfx/fx* (*Csf2rbΔDC*) mice were bred by crossing *Cd11cCre* and *Csf2rbfx* mice. Mice were bred and housed in specific pathogen-free conditions at the NIEHS with the following housing condition: light cycle: 7 AM to 7 PM, temperature: 72 ± 2 °F, humidity: 40 to 60%. Mice were euthanized with intraperitoneal injections of sodium pentobarbital (Vortech Pharmaceuticals) for lung cell isolation or analysis, or with $CO_2$ inhalation for LN, spleen, or bone marrow cell isolation. Male and female mice were used between 6 and 12 weeks of age.

### Allergic sensitization and a mouse model of asthma

For allergic sensitization, mice were lightly anesthetized with isoflurane and given one o.p aspiration of 100 μg LPS-free OVA (Worthington Biomedical) together with 10 μL HDE in a total volume of 50 μL in PBS (OVA/HDE)[20]. The HDE was prepared as previously described[63]. Briefly, vacuumed dust samples from homes in North Carolina were passed through a coarse sieve, then extracted at 100 mg/mL with PBS at 4 °C with overnight mild agitation. The samples were centrifuged to remove insoluble debris, and supernatants were sterilized by passage through a 0.22-μm filter (Millipore Sigma). Endotoxin concentration was 50 ng LPS/10 μL HDE, as measured by a Limulus Amebocyte Lysate assay (Lonza, Durham, NC). For the mouse model of allergic asthma, animals were lightly anesthetized with isoflurane and sensitized with two o.p. aspiration of OVA/HDE 7 days apart. In some experiments, mice received 1 μg PTX (Cayman, Ann Arbor, MI) in 50 μL PBS by o.p. 24 hours prior to allergic sensitization. Seven days after the second sensitization, mice were challenged by exposure to aerosolized 1% OVA in sterile PBS for 1 h. Mice were harvested 48 h post-challenge and BALF and lung tissue were collected. Lung tissues were incubated in 500 μL complete RPMI1640 containing 10% fetal bovine serum (Hyclone/Cytiva, West Sacramento, CA), penicillin/streptomycin, and 50 ng/mL β-mercaptoethanol (cRPMI-10) supplemented with 10 μg/mL OVA. To induce tolerance, the same procedure was followed except that seven days prior to the first sensitization, either 100 μg LPS-free OVA was given by o.p. aspiration, or 1 × 10⁵ cDCs or 1 × 10⁵ BMDCs were incubated with 10 nM OVA323-339 (New England Peptide) ex vivo for 1 h, washed, and adoptively transferred by o.p. aspiration.

### Isolation of cDCs from the lung and mLNs

Lungs were harvested from untreated mice or at 6 h, 16 h, or 18 h post-sensitization with OVA/HDE. Lungs were perfused by PBS injection into the right ventricle. For cDC preparation, lung tissue or mLNs were minced and digested with Liberase TM (100 μg/mL) (Roche), Collagenase XI (250 μg/mL), Hyaluronidase (1 mg/mL), and DNase I (200 μg/mL) (Sigma Aldrich) for 60 min. or 30 min. at 37 °C, respectively[63]. EDTA (20 mM final concentration) was added to stop the reaction. The digested tissue was then sieved through a 70 μm nylon strainer (BD Biosciences) to generate a single-cell suspension. To enrich for lung cDCs, low-density cells were collected by gradient centrifugation using 16% Nycodenz (Accurate Chemical), and then washed with PBS containing 0.5% bovine serum albumin and 2 mM EDTA. RBC lysis buffer was added to mLN DC preparation and then washed with PBS containing 0.5% bovine serum albumin and 2 mM EDTA. CD11c+I-A+CD88loF4/80loSiglec-FloLive/Dead− cDCs were purified by flow cytometric sorting.

### Adoptive transfer of alveolar macrophages

Alveolar macrophages were cultured with recombinant GM-CSF as described previously with some modifications[64]. Lungs from untreated C57BL/6J mice were digested for 30 min. as described above. After sieving through a 70 μm cell strainer, single-cell suspensions were washed, then cultured in cDMEM-10 containing 1x non-essential amino acids (ThermoFisher Scientific), 1x nucleosides (Millipore Sigma), and 10 ng/mL recombinant human GM-CSF (R&D Systems) in untreated petri dishes (BD Falcon). Seven days later, cells were harvested, passed through a 70 μm cell strainer, then cultured in new dishes with fresh media containing GM-CSF. Cultured cells were harvested in 1–2 weeks, then used for flow cytometric analyses or adoptive transfers. CD45+CD11b+ cells were 78% of total cells, and Siglec-F+ F4/80+ alveolar macrophages were over 98% of CD45+ cells. Cultured macrophages were adoptively transferred into *Csf2rbΔDC* mice. Mice were lightly anesthetized with isoflurane and macrophages were transferred by o.p. aspiration. Mice were allowed to rest for 4 weeks before use in subsequent experiments.

### Flow cytometric analysis and sorting

Cells were diluted to 1-2×10⁶/100 μL and incubated with a non-specific binding blocking reagent cocktail of anti-mouse CD16/CD32 Ab (2.4G2) (10% culture supernatant), 5% normal mouse and 5% rat serum (Jackson ImmunoResearch). Fluorochrome-conjugated antibodies (Abs) against cell surface antigens were obtained from BD Biosciences (BD), BioLegend (BL), R&D Systems (RD), Miltenyi Biotec (MB), Invitrogen/ThermoFisher (Inv), or eBioscience/ThermoFisher Scientific (eBio). The Abs included PE-anti-mouse CD3e (145-2C11, BL 100307; 1 μg/mL), BUV395-anti-mouse CD4 (RM4-4, BD 740209; 1 μg/mL), BUV395-anti-mouse CD11b (M1/70, BD 565553; 1 μg/mL), BV510-anti-mouse CD11b (M1/70, BL 101263; 1 μg/mL), PerCP-Cy5.5-anti-mouse CD11c (N418, BL 117328; 1 μg/mL), AF488-anti-mouse CD11c (N418, eBio 53-0114-82; 1 μg/mL), BV510-anti-mouse CD14 (Sa14-2, BL 123323; 1 μg/mL), APC eF780-anti-mouse CD14 (Sa14-2, BL 123331; 1 μg/mL), BUV395-anti-mouse CD24 (M1/69, BD 744471; 1 μg/mL), PE-Dazzle-anti-mouse CD24 (M1/69, BL 101837; 1 μg/mL), APC-anti-mouse CD25 (3C7, BL 101910; 1 μg/mL), FITC-anti-mouse CD40 (3/23, BD 561845; 1 μg/mL), BV510-anti-mouse CD44 (IM7, BL 103044; 1 μg/mL), FITC-anti-mouse CD45 (30-F11, BL 103108; 1 μg/mL), APC-Cy7-anti-mouse CD45.1 (A20, BL 110716; 1 μg/

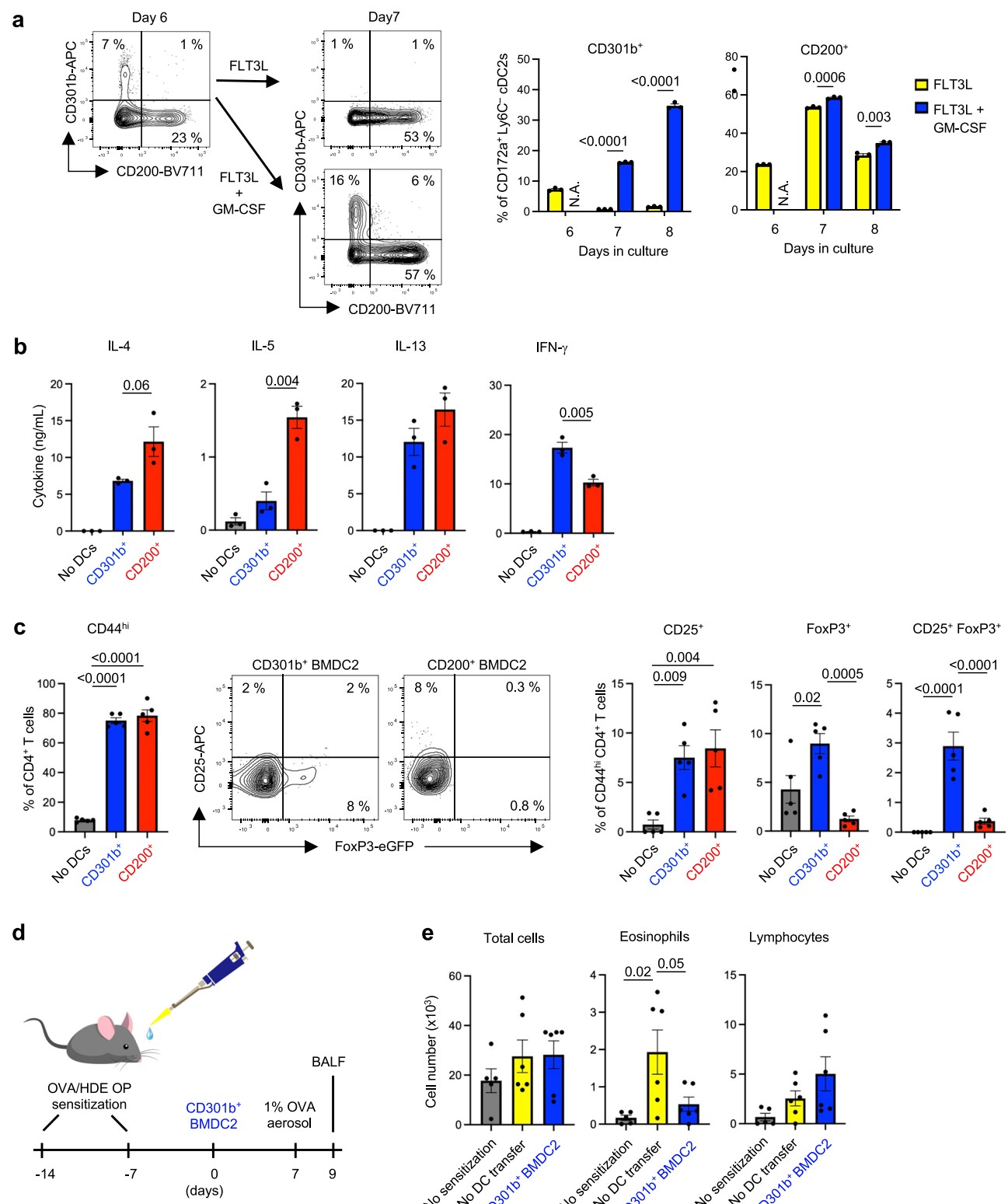

mL), BV510-anti-mouse CD45.2 (104, BL 109837; 1 µg/mL), PerCP-Cy5.5-anti-mouse CD45RB (C363-16A, BL 103313; 1 µg/mL), FITC-anti-mouse CD80 (16-10A1, eBio 11-0801-82; 1 µg/mL), BUV395-anti-mouse CD86 (P03, BD 745716; 1 µg/mL), FITC-anti-mouse CD86 (GL1, BD 561962; 1 µg/mL), PerCP-Cy5.5-anti-mouse CD88 (20/70, BL 135813; 1 µg/mL), PE-anti-mouse CD88 (20/70, BL 135806; 1 µg/mL), APC-anti-mouse CD88 (20/70, BL 135808; 1 µg/mL), BV711-anti-mouse CD88 (20/70, BD 743773; 1 µg/mL), BV510-anti-mouse CD103 (M290, BD 563087; 1 µg/mL), AF700-anti-mouse CD116/CSF2RA (698423, RD FAB6130N; 1 µg/

mL), PE-anti-mouse CD131/CSF2RB (REA193, BD 559920, 1 µg/mL), FITC-anti-mouse CD172a (P84, BD 560316; 2.5 µg/mL), BV711-anti-mouse CD200 (OX-90, BD 745548; 1 µg/mL), PE-anti-mouse CD200 (OX-90, BL 123807; 0.5 µg/mL), AF647-anti-mouse CD200 (OX-90, BL 123816; 1 µg/mL), PE-anti-mouse CD301b (URA-1, BL 146804; 1 µg/mL), APC-anti-mouse CD301b (URA-1, BL 146813; 1 µg/mL), BUV737-anti-mouse F4/80 (T45-2342, BD 749283; 1 µg/mL), PE-Dazzle594-anti-mouse F4/80 (BM8, BL 123146; 1 µg/mL), BV711-anti-mouse Ly-6A/E (D7, BL 108131; 0.5 µg/mL), BV510-anti-mouse Ly-6C (HK1.4, BL 128033;

**Fig. 7 | CD301b⁺ BMDC2s induce Tregs in vitro. a** BMDC2s displaying CD301b and CD200 on their surface. BMDC2s were generated by in vitro culture with FLT3L for 6 days, and further cultured with or without GM-CSF. Representative cytograms (left panels) and compiled data (right panels) of flow cytometric analysis are shown (*n* = 3 technical replicates). The gating strategy for BMDC2s (CD172a⁺CD11c⁺MHCII⁺CD24⁻Ly6C⁻CD88⁻Live/Dead⁻) is shown in Fig. S11a. **b** Induction of effector CD4⁺ T cells by BMDC2 subsets. Cytokine production from OT-II CD4⁺ T cells following culture with the indicated BMDC2 subsets was analyzed by ELISA (*n* = 3 technical replicates). Gating strategy is shown in Figure S11c. **c** Tregs in CD4⁺ T cells from *Foxp3*^eGFP OT-II mice were analyzed by flow cytometry after 5 days culture with BMDC2 subset (*n* = 5 technical replicates). Gating strategy is shown in Fig. S11d. **d** Timeline for mouse model of asthma to test tolerogenic function of BMDC2s. C57BL/6 mice were sensitized twice with OVA/HDE at days -14 and -7. CD301b⁺ BMDC2s were purified and incubated with OVA₃₂₃₋₃₃₉ peptides, then adoptively transferred by o.p. aspiration to sensitized mice on day 0.

Seven days post-transfer mice were challenged with OVA aerosol. **e** Cells in BALF were analyzed 48 hours after challenge. Cell numbers of the indicated leukocytes in BALF (No sensitization *n* = 5, No DC transfer *n* = 6, CD301b⁺ BMDC2s *n* = 6 biological replicates). **a** Represented results from three experiments. Data were analyzed by two-stage step-up unpaired *t*-test. **b**, **c**, **e** Represented results from two experiments are shown. **b** Data were analyzed by two-tailed unpaired *t*-tests. **c**, **e** Data were analyzed by one-way ANOVA with Tukey's multiple comparison test. Each dot represents separately cultured cells (**a**–**c**) or individual mouse (**e**). Data are presented as mean values ± SEM. *P* values are indicated above the graphs. Source data are provided as a Source Data file. BMDC bone marrow dendritic cells, Treg regulatory CD4⁺ T cells, FLT3L FMS-like tyrosine kinase 3 ligand, GM-CSF granulocyte macrophage colony stimulating factor, ELISA enzyme-linked immunosorbent assay, N.A. not applicable, OVA ovalbumin, HDE house dust extract, OP oropharyngeal, BALF bronchoalveolar lavage fluid.

1 µg/mL), FITC-anti-mouse Ly-6C (AL-21, BD 553104; 2.5 µg/mL), APC eFluor 780-anti-mouse Ly-6C (HK1.4, eBio 47-5932-82; 1 µg/mL) eFluor450-anti-mouse MHC class-II I-A^b (AF6-120.1, eBio 48-5320-82; 1 µg/mL), FITC-rat IgG2a (R35-95, BD 554688; 1 µg/mL), PE-anti-mouse Siglec-F (S17007L, BD 552126; 0.5 µg/mL), BV711-anti- Siglec-F (E50-2440, BD 740784; 0.25 µg/mL), AF647-anti- Siglec-F (E50-2440, BD 562680; 1 µg/mL), APC-rat IgG2a (eBR2a, eBio 17-4321-81; 1 µg/mL), BV711-rat IgG2a (RTK2758, BL 400551; 1 µg/mL), PE-rat IgG2a (RTK2758, BL 400508; 0.5 µg/mL), PE-rat IgG2b (eB149/10H5, eBio 12-4031-82; 0.5 µg/mL), FITC-rat IgMκ (RTK2118, BL 400805; 2.5 µg/mL). For transcription factor analysis, cells were permeabilized using Foxp3/Transcription Factor Staining Buffer Set (eBio 00-5523-00), then stained with the following Abs; eFluor450-anti-mouse Foxp3 (FJK-16s, eBio 48-5773-82; 2 µg/mL), APC-anti-GATA3 (W19195B, BL 386908; 0.5 µg/mL), PE-anti-mouse HELIOS (22F6, BL 137206; 1 µg/mL), BV650-anti-mouse RORγt (Q31-378, BD 564722; 0.5 µg/mL µg/mL), eFluor450-rat IgG2aκ (eBR2a, eBio 48-4321-82; 2 µg/mL), APC-rat IgG2aκ (RTK2758, BL 400511; 0.5 µg/mL), PE-anti-hamster IgG (HTK888, BL 400907; 1 µg/mL), BV650-anti-mouse IgG₁κ (R19-15, BD 744532; 0.5 µg/mL). Stained cells were analyzed on an LSR-Fortessa flow cytometer (BD Biosciences), and the data were analyzed using FACS Diva (BD Biosciences) and Cytobank (Cytobank) or FlowJo (Treestar) software. Only single cells were analyzed or purified, and dead cells stained with eFluor780-conjugated Live/Dead dye (ThermoFisher Scientific) were excluded from analysis. For purification, stained cells were sorted using a cell sorter, FACS ARIA-II (BD Biosciences). Antibodies used are listed in Supplementary Table 1. The gating strategies are depicted in Supplementary Figures.

### Coculture of cDCs and CD4⁺ T cells

To isolate CD4⁺ T cells from skin-draining LNs and spleens, single cell suspensions were passed through a 70 µm strainer and T cells were enriched by gradient centrifugation using Histopaque 1083 (Millipore Sigma). Naïve CD4⁺ T cells were purified by AutoMACS (depl025 program, Miltenyi) using streptavidin-conjugated MACS beads (Miltenyi) and a biotinylated antibody cocktail containing the following Abs; anti-mouse CD8α (53-6.7, BD 553029; 0.5 µg/mL), CD8β (53-5.8, BD 553039; 0.5 µg/mL), CD11b (M1/70, BD 553309; 0.5 µg/mL), CD11c (HL3, BL 117304; 0.5 µg/mL), CD16/32 (2.4G2, BD 553143; 0.5 µg/mL), CD19 (6D5, BL 115504; 0.5 µg/mL), CD25 (PC61, BL 102004; 0.5 µg/mL), CD44 (IM7, BL 103004; 0.05 µg/mL), CD49b (DX5, BD 553856; 0.5 µg/mL), I-A^b (AF6.120.1, BL 116404; 0.5 µg/mL) and Ly-6C/G (RB6-8C5, BD 553125; 0.5 µg/mL). Antibodies used are listed in Supplementary Table 1. Naïve CD4⁺ T cells (5×10⁴ cells/well) and cDCs (5×10³ cells/well) were cocultured for 5 days in 200 µL complete Iscove's modified Dulbecco's medium (IMDM) containing 10% fetal bovine serum (FBS; certified, Invitrogen), 50 µM β-mercaptoethanol, penicillin, and streptomycin in a 96-well U-bottom plate (BD Biosciences) in a CO₂ incubator. In some experiments, inhibitors of TGF-β signaling were

added to the culture at the indicated concentrations: SB431542 (TOCRIS, Bristol, UK), SD 208 (TOCRIS), RepSOX (TOCRIS), Disulfram (TOCRIS), and anti-EBI3 Ab (clone V1.4C4.22, Millipore Sigma). For cytokine analysis, the cells were harvested and washed 5 days after coculture, and viable cells were counted using Luna-FL cell counter (Logos Biosystems). T cells were then incubated (1 × 10⁵ cells/200 µL/well) for 24 h in a 96-well flat-bottom plate coated with antibodies to mouse CD3ε (145-2C11, BL 100331; 1 µg/mL) and CD28 (37.51, BL 102116; 1 µg/mL). Cytokines in the supernatant of incubated T cells were measured by ELISA using Multiskan Ascent plate reader with Ascent 2.6 software (Thermo Electron) according to the manufacturer's instructions.

### Bone marrow-derived cDC culture

Bone marrow cells were prepared from femurs, tibiae, humeri and sternums of C57BL/6 or *Csf2rb*^ADC mice, and red blood cells were lysed with 0.15 M ammonium chloride and 1 mM potassium bicarbonate. Bone marrow cells were then cultured in complete RPMI1640 containing 10% fetal bovine serum (Hyclone/Cytiva, West Sacramento, CA), penicillin/streptomycin, and 50 ng/mL β-mercaptoethanol (cRPMI-10) supplemented with 100 ng/mL recombinant human FLT3L (rFLT3L) (NIEHS Protein Expression Core) at 2 × 10⁶/mL for 6 days[63]. Half of the media was replaced with fresh media on day 3 of culture. The cells were cultured for an additional 2 days with or without with 10 ng/mL rmGM-CSF (R&D Systems, Minneapolis, MN) in the presence of rFLT3L. Surface display of CD200 and CD301b on cultured BMDC2s (CD11c⁺I-A⁺CD26⁺CD172a⁺CD24⁻CD88⁻Live/Dead⁻) were analyzed by flow cytometry.

### In vivo cDC maturation assay

CD200⁻CD301b⁻Ly6C⁺ cDC2s or CD200⁻CD301b⁺Ly6C⁻ cDC2s from the lungs of OVA/HDE-sensitized C57BL/6 mice (CD45.2) were purified by flow cytometry and adoptively transferred into syngeneic CD45.1 mice by o.p. aspiration (0.5–1.5 × 10⁵ cells/recipient). CD45.2⁺ donor-derived cDC2s in the recipient lungs were analyzed by flow cytometry.

### Lung cDC migration assay

Mice were given PKH26 (10 µM) (Sigma) with or without OVA/HDE by o.p. aspiration (50 µL/mouse). Some mice received 1 µg PTX by o.p. aspiration 24 h prior to the PKH26 aspiration. MLNs were harvested 24 h after treatment, and digested for 30 min. Single cells were stained with Abs after RBC lysis. PKH26⁺ migratory cDC2s were analyzed by flow cytometry.

### Analysis of Treg generation in the lung

Naïve CD4⁺ T cells were purified from CD45.1 OT-II mice as described above, and 10⁶ cells were intravenously injected through the tail veins. Some recipient mice were then given OVA alone (100 µg) by o.p.

aspiration or PTX (1 µg) 24 h prior to the OVA aspiration. Lungs of recipient mice were harvested 5 days later, and digested for 30 min. T cells were enriched by gradient centrifugation using Histopaque 1083. Surface proteins on and intracellular Foxp3 in CD45.1⁺ donor CD4⁺ T cells were analyzed by flow cytometry.

### Imaging of precision-cut lung slices (PCLS)

PCLSs were generated, stained, and visualized as previously described[65]. Briefly, slices of the right superior lobe of fresh lungs inflated with 2% agarose (GeneMate Sieve GQA) were made using a precision-cut tissue slicer VF-300 Compresstome (Precisionary Instruments) at 110 µm thickness without perfusion. The slices were stained with AF488-anti-CD324 (DECMA-1, Inv 560061, 2.5 µg/mL), BV605-anti-CD11c (N418, BL 117334; 1 µg/mL), PE-anti-F4/80 (BM8, BL 123110; 1 µg/mL), and APC-anti-CD103 (2E7, BL 121414; 1 µg/mL) or APC-anti-CD301b (URA-1, BL 146814; 1 µg/mL). Stained slices were fixed, then analyzed using a multi-photon laser-scanning microscope Zeiss 980 (Carl Zeiss) and Zen software (Bitplane). Antibodies used are listed in Supplementary Table 1.

### Cellular indexing of transcriptomes and epitopes sequencing (CITE-Seq)

CD11b⁺ cDC2s from OVA/HDE-treated or untreated C57BL/6 mice were purified by flow cytometry and stained with Total A-Seq oligo-conjugated Abs against CD200 (OX-90, BL 123811), CD301b (URA-1, BL 146817) and Ly6C (HK1.4, BL 128047) according to the manufacturer's instruction (BioLegend) (https://www.biolegend.com/en-us/totalseq)[66]. Antibodies used are listed in Supplementary Table 1. The cells were counted and examined for viability using a TC-20 cell counter (Bio-Rad). Approximately 10,000 live cells at or above $3 \times 10^5$ cells/mL with 90% or higher viability were loaded into the Single Cell Chip, followed by forming a single cell emulsion in Chromium Controller (10x Genomics). The cDNA for antibody-derived transcripts (ADT) and gene-derived transcripts was generated and amplified according to the manufacturer's instructions (10x Genomics and BioLegend). The Gene Expression library and the ADT Library were prepared using the Chromium single cell 3′ library and gel bead kit v3 (10x Genomics, catalogue #PN-1000073) and additional reagents recommended in the protocol of Total A-seq (BioLegend). The two libraries were mixed at a 10:1 molar ratio (Gene Expression library to ADT library) and sequenced by the NIEHS Epigenomics and DNA Sequencing Core Laboratory on NovaSeq 6000 (Illumina) with paired-end sequencing. The data were processed using RTA version 2.4.11. A total of $2.4 \times 10^9$ reads were obtained.

### Analysis of CITE-seq data

CITE-Seq raw data processing: Alignment, barcode assignment and unique molecular identifier (UMI) counting was performed using Cell Ranger 3.1.0 and the cellranger count command. Alignment was performed with the mouse mm10-1.2.0 reference. The following feature libraries were included for antibody sequencing: CD200 (TCAATTCCGGTAGTC), CD301b (CTTGCCTTGCGATTT), CD11b (TGAAGGCTCATTTGT), and Ly6C (AAGTCGTGAGGCATG). Outputs from filtered count matrices were used for subsequent analyses. Rounded estimates from Cell Ranger were 3,500/5,000/6,500 cells, 195,000/120,000/90,000 mean reads per cells, and 3,000/3,100/3,500 median genes per cell for the 0 h, 6 h, and 18 h samples, respectively. 96% of RNA reads mapped to the reference genome, and 97% of all barcodes were valid.

scRNA-Seq dimensionality reduction and clustering: Data from scRNA-Seq were processed using the Seurat v3.0 package in R version 3.6.1 (http://satijalab.org/seurat/)[67]. Data were filtered on characteristics for homogeneity, including number of features (high threshold: 6000; low threshold: 1000), total RNA counts (high threshold: 50,000; low threshold: 250), proportion cycling (high threshold: 0.08; low threshold: 0.02), and proportion of mitochondrial RNA (high threshold: 0.075; low threshold: 0.005). Data were normalized and scaled for the number of RNA features, proportion cycling, and proportion of mitochondrial RNA. Normalized and scaled gene expression data were projected onto principal components (PCs). The first 30 PCs were used for non-linear dimensionality reduction using UMAP[68]. Gene expression and metadata were visualized using this UMAP projection. Clustering was performed using the FindNeighbors (k.param = 50), followed by the FindClusters (resolution = 0.5) functions of the Seurat v3.0 package in R version 3.6.2[67]. Cluster marker genes from res.0.5 in Seurat (described above) were generated by the FindAllMarkers function. Spliced and unspliced counts matrices were constructed with the velocyto run10x function[30], followed by RNA velocity analysis via scVelo version 0.2.4[69], according to the authors' tutorials at https://scvelo.readthedocs.io/en/stable/. The results are displayed as velocity stream and latent time views with scVelo's dynamical mode. CITE-Seq data have been deposited in GEO (GSE261034). Expression of the mouse *Csf2* gene in the lung was analyzed using a scRNA-Seq dataset previously published by Han et al.[35], and visualized by UMAP projection.

### Statistics

Data are presented as mean ± SEM. Statistics to analyze differences among groups using Prism software are indicated in figure legends. Outliers identified by GraphPad Prism ROUT method (Q = 1%) were removed from analysis. $P < 0.05$ was considered significant.

### Reporting summary

Further information on research design is available in the Nature Portfolio Reporting Summary linked to this article.

## Data availability

CD11b⁺ cDC2 CITE-Seq data generated in this study have been deposited in GEO under accession code GSE261034. Other scRNA-Seq (GEO accession code GSE1080974) and bulk RNA-Seq (GEO accession code GSE149778) data from previously published papers were also used in this study[18,35]. All other data are available in the article and its Supplementary files or from the corresponding author upon request. Source data are submitted to figshare: https://doi.org/10.6084/m9.figshare.28596968. Source data are provided with this paper.

## Code availability

Custom codes were not created for data analyses in this study. Analysis followed publicly available instructions from Seurat (http://satijalab.org/seurat/) and scVelo (https://scvelo.readthedocs.io/en/stable/). Code for figure reproducibility is available in Supplementary Data files in Supplementary Information.

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

## Acknowledgements

We thank Burkhard Becher (University of Zürich, Switzerland) for distribution of *Csf2rb^{fx}* mice, Alessandra Livraghi-Butrico (University of North Carolina) for help with histology, Xin Xu, Jason Malphurs and Brian Papas for help with CITE-Seq, Gentaro Izumi, Maria Sifre and Carl Bortner for help with flow cytometry and cell sorting, Ligon Perrow for help with mouse colony management, and Prashant Rai and Stavros Garantziotis (NIEHS) for critical reading of the manuscript. This work was supported by the Intramural Research Program of the National Institutes of Health, the National Institute of Environmental Health Sciences (ZIA ES102025-09) to DNC.

## Author contributions

H.N. and D.N.C. conceived of the project. H.N. and C.L.W. designed experiments. C.L.W., H.N., K.N. and G.S.W. performed experiments and analyzed data. S.A.G., P.W.K., and M.B.F. analyzed CITE-Seq data. Y.A. and P.J.B. generated *Csf2^{ΔARE}* mice. H.N., C.L.W. and D.N.C. wrote the manuscript. All authors contributed to the discussion and review of the manuscript.

## Funding

## Competing interests

The authors declare no competing interests.
