## [Transparent Peer Review file · Nature Communications]

GM-CSF-dependent CD301b⁺ mouse lung dendritic cells confer tolerance to inhaled allergens

Corresponding Author: Dr Hideki Nakano

Version 0:

Reviewer comments:

Reviewer #1

(Remarks to the Author)

This manuscript from Wilkinson et al describes a subset of cDC2 cells in the lung that promote the generation of Tregs during homeostasis and differentiate into Th2-promoting APCs during inflammatory responses. While that data consistent with their overall hypothesis, there are aspects of the manuscript that need to be dealt with before a decision on publication can be made

1. The title needs to be changed-while they showed a decrease in the CD301b⁺ cDC2s in CD11c-Cre/Csfrb⁰flox mice, they do not show any data that GM-CSF is acting directly on that subset. CD11c-Cre will delete in not only DCs but also several subsets of interstitial macrophages as well at alveolar macrophages, so the effect could be indirect. They need to specifically show that GM-CSF signalling in the CD301b⁺ subset is important

2. The authors conclude that TGFb1 from the CD301b⁺ cDC2s is critical for Treg generation. This is based solely on gene expression studies. The authors need to show this directly in order to make such a sweeping claim. Tgfb1 conditional KO mice are available for this work.

Reviewer #2

(Remarks to the Author)

This paper deals with the important issue whether CD301b positive cDC2s are promoting Th2 immunity (as has been stated by a few groups). The paper shows that this is clearly not the case in the lungs, but rather proposes that CD301b⁺ cDC2 are mediating tolerance. Strikingly, these cells seem to depend on GM-CSF. The CD301b⁺ cDC2 induce tolerance via expansion of Foxp3⁺ Tregs in a TGFb dependent manner.

The Th2 inducing capacity of cDC2s seems to reside in CD200 positive cDC2s, that develop from CD301b + cDC2, which might explain why others found a link between CD301b expression and capacity to induce Th2.

Comments :

General :

Given that DC3 have been proposed to depend on GM-CSF, and the authors propose an important role for GM-CSF; how are these CD301b⁺ cDC2 expressing Ly6C related to cDC3. Are they derived from Ly6Chi DC3 progenitors ?

My major critique is that all the functional and mechanistic data on induction of Tregs by CD301b⁺ cDC2 is done ex vivo using sorted cDC2 subsets and OT2 T cells. There is also some adoptive transfer experiments done, but these do not exclude the transfer of injected antigen or apoptotic cells to endogenous APCs. Also the use of Ccr7 deficient mice to make a point about migratory DC function is problematic since the anatomy of the immune system in these mice is totally abnormal. Use of a pertussis toxin could be an additional proof.

The authors could also have used Mgl2Cre or DTR mice to make their point in vivo.

Specific comments about figures :

In Figure 1f, the authors should also show if adoptive transfer of OVA carrying CD200+ cDC2 boosts Th2 immunity, as predicted from the ex vivo experiment with OT2 cells.

Fig 2c is unclear. Which color represents what? Is this latent time? Pseudotime? Both combined and individual plots seem to show that cluster 4 & 8 are end states, so how can they give rise to cluster 2 & 5, which also seem end states? The same can be stated for cluster 2 which is stated to be a state before cluster 5, yet they share the same latent time/pseudotime.

175-186 : but why do they lose CD301b? There is no inflammation. The scdata suggests that only during OVA/HDE the CD301b should go down? Maybe there is insufficient "space" or "niche" and associated factors for these CD301b DCs to retain their phenotype?

Fig 3c: what does the Y-axis mean? What subsets? Is the "fold change" frequency or cell counts?

245-247 : why would these DCs be in the alveolar ducts and not around the larger airways? Data on larger airways are not provided. Maybe there are also CD301b DCs there, who sense Csf2 from ILC2s instead?

250-252 Itgax-cre x Csf2rb : CD11c-cre targets the entire myeloid fraction, so all myeloid cells will lack Csf2rb. Alveolar macrophages are highly disturbed in this model. Maybe an inflammatory setting is created due to continuous AM turnover (death & monocyte influx) which causes the decrease in CD301b DCs? In my view, a Zbtb46-Cre would be better if available? In any case, this downside of the model should be clearly stated.

265-274 : To account for the issue mentioned above this experiment (culturing of DCs lacking Csf2rb) should be done with BM DCs maybe or DCs from skin? Right now the ENTIRE cDC2 fraction is used, but this fraction has already been influenced by the disturbed environment due to AM turnover.

Reviewer #3

(Remarks to the Author)

Manuscript number: NCOMMS-24-30321-T

Title: "GM-CSF-dependent CD301b+ lung dendritic cells confer 1 tolerance to inhaled allergens" by Christina L. Wilkinson et al.

In the present manuscript by Christina L. Wilkinson et al., the authors intended to identify dendritic cell (DC) subpopulations that are responsible for CD4+FOXP3+ regulatory T (Treg) cell induction in mouse lungs. Using OVA-sensitization and preclinical mouse models of allergic asthma, the authors identify CD301b+ DCs that are comparatively strong in inducing Treg cell development or expansion. Employing conditional knock out mouse models the authors demonstrate that development/Expansion of CD301b+ DCs depends on GM-CSF. Further, the authors show that under inflammatory conditions CD301b+ DCs can differentiate into CD200+ DCs promoting Type 2 immunity and TH2 differentiation and that transfer of CD301b+ T cells might offer potential for immunological tolerance induction to inhaled allergens.

Validity

Data presented in this paper are thoroughly analyzed and statistical testing is performed where indicated. Data are not over-interpreted, although a few logical inconsistencies have been detected and improvements are suggested below. In general, there is no concern about the validity of the findings.

This is a well-written manuscript providing strong evidence for a role of CD301b+ DCs in Treg cell induction in the lungs of mice. However, several concerns rose during the review:

Analytical approach

No concerns.

Major critique

1. Based on the expression of HELIOS, RORgt, GATA-3 and other transcription factors, CD4+FOXP3+ regulatory T cells can be subdivided and better characterized. Since the authors emphasize in this manuscript that CD301b+ DCs influence the development of peripheral (p)Treg cells, the authors should be motivated to better characterize CD301b+ DC-induced Treg cells at the level of transcription factor expression. In addition, the authors should measure ability of the different DC subsets described in this manuscript for their capacity to induce proliferation of different Treg cell subsets.
2. Next to demarcating peripheral Treg cells from Treg cells of thymic origin, subsets have been described based on their capacity to suppress and/or to promote tissue repair. Given the role of tissue repair Treg cells in type 2 immunity, the authors should be encouraged to functionally phenotype the Treg cells induced by CD301b+ DCs.
3. In figures 1c and d, 3c, 5e, 6c and 7c the authors test the ability of different DC subsets from wt and transgenic mice to induce FOXP3 expression in CD4+ T cells. What's the fate of the FOXP3-negative CD4+ T cells under these various conditions in terms of other T cell lineage determining transcription factors like GATA-3, T-bet, RORgt? In the same vein and as previously asked, it would be interesting to know which transcription factors the FOXP3+ Treg cells still express (HELIOS, GATA-3, RORgt).
4. In the same sense, in figure 1f the authors analyzed total cell count, eosinophils, neutrophils and lymphocyte upon transfer of CD301b+ and CD301b- DC prior to OVA/HDE sensitization. How about expression of FOXP3, GATA-3, T-bet and RORgt

in CD4+ T cells under these conditions?

5. In previous reports it was shown that Klf4 (Tussiwand et al. Immunity 2015) or PD-1 and IRF-4 expressing DC are inducers of type 2 immunity (Williams et al. Nat Comm 2013, Gao et al. Immunity 2013). How can CD200+ DCs be differentiated from this Klf4 or IRF-4 and PD-1 expressing DC subset and what is the expression of these molecules in CD301b+ DC?

6. In figure 6 the authors nicely demonstrate the role of CD301b+ DC-produced TGF-beta in the induction of Treg cells. Since Treg cell induction relies on TGF-b and IL-2 it would be interesting to understand the source of IL-2 in this context. In the same vein, next to Treg cell differentiation TH17 cell differentiation can also be promoted by TGF-b in context with IL-6. Do CD301b+ DCs also promote TH17 cell differentiation?

7. In figure 7 the authors demonstrate that adoptive transfer of CD301b+ DCs can confer immunological tolerance. In their introduction to authors refer to SCIT and SLIT as treatment possibilities to induce allergen-specific tolerance. Does SCIT and/or SLIT induce expansion/ de novo generation of CD301b+ DC via GM-CSF in humans?

Minor critique

8. Given the role of CD301b+ DC in Treg cell induction I was wondering whether Csf2deltaARE mice harbor higher Treg cell numbers already under steady state conditions in the lungs?

Version 1:

Reviewer comments:

Reviewer #1

(Remarks to the Author)

The authors have addressed the issues raised in the original submission and the manuscript is now acceptable for publication.

Reviewer #3

(Remarks to the Author)

In the revised manuscript by Hideki Nakano and colleagues the authors have satisfactorily responded to most of my major criticisms with inclusion of additional data further substantiating the important role of CD301b+ DC in induction of Treg cells. Most importantly, the authors have now provided new data further characterizing the induced Treg cells and CD301b+ DC in terms of IRF4, KLF4, PD-1 and IL-2 expression.

Overall the manuscript is greatly improved and I would like to recommend it for publication in Nature Communications.

Reviewer #1

This manuscript from Wilkinson et al describes a subset of cDC2 cells in the lung that promote the generation of Tregs during homeostasis and differentiate into Th2-promoting APCs during inflammatory responses. While that data consistent with their overall hypothesis, there are aspects of the manuscript that need to be dealt with before a decision on publication can be made.

Response. Thank you for careful reading and insightful comments. We added new results and descriptions to address reviewer's concerns. Please see our point-by-point responses shown below. The revised descriptions along with additional results are highlighted by yellow color in the main text, and relevant references are listed at the end of this response document.

Comment 1. *The title needs to be changed-while they showed a decrease in the CD301b+ cDC2s in CD11c-Cre/Csf2rb0flox mice, they do not show any data that GM-CSF is acting directly on that subset. CD11c-Cre will delete in not only DCs but also several subsets of interstitial macrophages as well at alveolar macrophages, so the effect could be indirect. They need to specifically show that GM-CSF signalling in the CD301b+ subset is important.*

Response. Thank you for the insightful comment. As this reviewer pointed out, macrophage numbers are decreased in lungs of *Csf2rb^{ADC}* (*Itgax^{Cre} x Csf2rb^{flx}*) mice, and this could potentially influence cDC numbers through an indirect pathway. As shown in the original submission (Figure 7a), the addition of GM-CSF to cultured BMDC2s increased CD301b⁺ BMDC2s (Line 377-383). We believe this is a direct effect of GM-CSF on DC2s, as there are very few macrophages in these cultures with FLT3L. Nevertheless, we addressed this point in an *in vivo* experiment by performing adoptive transfers of alveolar macrophages (AMs) to *Csf2rb^{ADC}*. This was done because a previous study reported that adoptive transfer of wild-type macrophages to GM-CSF receptor-deficient mice rescues the number of lung macrophages and suppresses proteinosis¹. In our experiment, *Csf2rb^{ADC}* mice that received AMs had similar numbers of lung macrophages as *Csf2rb^{flx}* control mice, whereas CD301b⁺ cDC2 numbers were still significantly lower in *Csf2rb^{ADC}* mice. These new results are presented in Extended Data Figure 8b and 8c, and a description was added to the main text (Line 304-311 and 322-325) and Methods were added (Line 689-701) in revision. Together, these experiments support our conclusion that GM-CSF is important for the development of CD301b⁺ DC2s, and we have consequently not changed the title.

Comment 2. *The authors conclude that Tgfb1 from the CD301b+ cDC2s is critical for Treg generation. This is based solely on gene expression studies. The authors need to show this directly in order to make such a sweeping claim. Tgfb1 conditional KO mice are available for this work.*

Response. As this reviewer stated, Figure 6b and Extended Figures 9b and 9c (Extended Figures 6c and 6d in original submission) show *Tgfb1* expression in CD301b⁺ DCs. However, our conclusion that TGF- β is important for Treg induction is not based solely on this observation. In our original submission, we showed that the TGF- β receptor inhibitor, SB431542, which inhibits ALK5/TGF- β RI serin/threonine kinase-transducing signals from TGF- β R suppresses Treg differentiation in a dose-dependent manner (Figure 6c). For further confirmation of the specific requirement of TGF- β signaling for Treg differentiation, we tested two additional TGF- β inhibitors; 1) SD208, a potent ATP-competitive TGF- β RI inhibitor; and 2) RepSOX, another ALK5 inhibitor. Both of these inhibitors suppressed Treg differentiation when they were added to co-culture of naïve CD4⁺ T cells and lung cDC2s. These new results are now presented in Figure 6d and Extended Data Figure 10d, and are described in the Results section (Line 364-366). The Methods section now includes the use of these two new inhibitors (Line 767-769).

Reviewer #2

This paper deals with the important issue whether CD301b positive cDC2s are promoting Th2 immunity (as has been stated by a few groups). The paper shows that this is clearly not the case in the lungs, but rather proposes that CD301b+ cDC2 are mediating tolerance. Strikingly, these cells seem to depend on GM-CSF. The CD301b+ cDC2 induce tolerance via expansion of Foxp3+ Tregs in a TGFb dependent manner. The Th2 inducing capacity of cDC2s seems to reside in CD200 positive cDC2s, that develop from CD301b + cDC2, which might explain why others found a link between CD301b expression and capacity to induce Th2.

Response. Thank you for careful reading and productive comments. To address reviewer's concerns, we performed experiments, and added new results and descriptions to the manuscript. Also, some results are shown in this response. Please see our point-by-point responses shown below. The revised descriptions along with additional results are highlighted by yellow color in the main text, and relevant references are listed at the end of this response document.

General :

Comment 1. *Given that DC3 have been proposed to depend on GM-CSF, and the authors propose an important role for GM-CSF; how are these CD301b+ cDC2 expressing Ly6C related to cDC3. Are they derived from Ly6Chi DC3 progenitors ?*

Response. Thank you for the insightful comment. As this reviewer mentioned, Bourdely *et al.* previously reported that human CD88⁻CD14⁺CD1c⁺CD163⁺ cells are DC3s that arise independent of CDPs or cMoPs, and that their development is dependent on GM-CSF². CD88⁻CD14⁺CD1c⁺CD163⁺ cells were annotated as DC3 by transcriptomic comparison of human cells in Bourdely's study with previously published scRNA-Seq data from Villani's human cell study³. To investigate the relationship between mouse lung cDC2 subsets and human DC3, we compared mouse lung cDC2 transcriptomic data from the present study with the transcriptome of human DC3 from Villani's scRNA-Seq data. As depicted in Extended Data Figure 3e, cluster 7, which is one of the Ly6C⁺ populations in mouse lung cDC2s, was closely related with human DC3. A dendrogram displaying the relationships among mouse cDC2 clusters was generated using the Seurat program and indicates that CD301b⁺ cDC2 clusters 4 and 8 are more distantly related to Ly6C⁺ clusters, including cluster 7 (Extended Data Figure 3d). These results suggest that CD301b⁺ cDC2s are not closely related to DC3 or DC3 progenitors. We have added the above analyses to Extended Data Fig. 3d and 3e of our revised manuscript and also added related information to the Results section (Line 174-180).

In addition, as we have shown the present paper (Figure 2h and 2i) and in a previous paper⁴, CD301b⁺ cDC2s give rise to CD200⁺ cDC2s, indicating that CD301b⁺ cDC2s are in the same developmental lineage as CD200⁺ cDC2s, further suggesting that CD301b⁺ cDC2 are not DC3.

Csf2rb^{ADC} mice had normal numbers of lung Ly6C⁺ cDC2s as shown in Figure 5b, suggesting that majority of Ly6C⁺ cDC2s arise independently of GM-CSF. It is possible that cluster 7 in Ly6C⁺ cDC2s might be a mouse counterpart of human DC3, but we feel that the development and function of Ly6C⁺ cDC2s are outside the scope of the present study because Ly6C⁺ cDC2s are not the main DC subset stimulating Treg differentiation. Therefore, we did not pursue the development of Ly6C⁺ cDC2 clusters in the present study.

Comment 2. *My major critique is that all the functional and mechanistic data on induction of Tregs by CD301b+ cDC2 is done ex vivo using sorted cDC2 subsets and OT2 T cells. There is also some adoptive transfer experiments done, but these do not exclude the transfer of injected antigen or apoptotic cells to endogenous APCs. Also the use of Ccr7 deficient mice to make a point about migratory DC function is problematic since the anatomy of the immune system in these mice is totally*

abnormal. Use of a pertussis toxin could be an additional proof. The authors could also have used *Mgl2Cre* or *DTR* mice to make their point *in vivo*.

Response. Thank you for the insightful comments. As this reviewer mentioned, we tested the function of multiple cDC subsets, including cDC1 and cDC2 subsets in *ex vivo* experiments, and found that Tregs were selectively and efficiently induced by CD301b⁺ cDC2s (Figure 1c, d, 4g, 5e, and Extended Data Fig. 8d). We also tested the *in vivo* function of cDCs by adoptive transfer of CD301b⁺ and CD301b⁻ cDC2s, and found that CD301b⁺ cDC2s, but not CD301b⁻ cDC2s, induce tolerance *in vivo* (Figure 1e and f). Although we have not presented direct evidence of Treg induction by CD301b⁺ cDCs *in vivo*, subset-specific function suggests that Tregs are not induced by bystander cells in a non-specific manner. Also, we found selective expression of *Tgfb1*, *Furin* and *Nrros* by CD301b⁺ cDC2 subsets (Figure 6b), and demonstrated that TGF-β is required for Treg induction by lung cDC2s (Figure 6c, d and Extended Data Figure 10d). We believe our experiments are well controlled using other cDC subsets, and we demonstrate that Tregs are specifically induced by CD301b⁺ cDC2, and present molecular mechanism for Treg induction.

This Reviewer proposed that we employ *Mgl2^{Cre}* or *Mgl2^{DTR}* mice to test the function of CD301b⁺ cDC2s. However, Macrophage Galactose-type C-type Lectin 2 (MGL2) is also produced by macrophages, as demonstrated by the original report⁵, in addition to DCs. In agreement with that previous report and with ImmGen gene expression database (<http://rstats.immgen.org/Skyline/skyline.html>), we found that *Mgl2* is expressed by interstitial lung macrophages in addition to cDC2s (see accompanying graph). Therefore, ablation of *Mgl2*-expressing cells using *Mgl2^{Cre}* x *Dta^{flx}* or *Mgl2^{DTR}* mice would not selectively target CD301b⁺ cDC2s. Thus, we studied the cellular functions of purified cDC2s in *ex vivo* experiments and in adoptive transfer models.

As this reviewer correctly pointed out, *Ccr7*-deficient mice have multiple defects in immune cells and immune responses. Thus, our original data supported, but did not conclusively prove, that DC migration from the lung to LNs is dispensable for Treg development. We therefore followed this reviewer's advice and tested Treg generation in the lung after administration of pertussis toxin (PTX) to block cDC migration. Our experiments showed that although PTX efficiently blocked cDC migration from the lung to draining LNs (Extended Data Fig. 5b), OVA inhalation without adjuvants induced Tregs in PTX-treatment mice at a comparable level with PTX-untreated mice (Fig. 3c). The new results are presented in Extended Data Fig. 5b and Figure 3c, and descriptions along with the new results were added to the Result section (Line 223-230), the Discussion (Line 463-465), and the Methods (Line 664-665, 797, 803-804).

Specific comments about figures :

Comment 3. In Figure 1f, the authors should also show if adoptive transfer of OVA carrying CD200⁺ cDC2 boosts Th2 immunity, as predicted from the *ex vivo* experiment with OT2 cells.

Response. This is an excellent suggestion. To comply with it, we sensitized mice with OVA/HDE, isolated CD200⁺ cDC2s from the lungs of these animals and adoptively transferred these cells into lungs of naïve mice. These recipient animals were then challenged with OVA to test the effect of CD200⁺ cDC2s *in vivo*. As reviewer anticipated, adoptive transfer of OVA-bearing CD200⁺ cDC2s enhanced eosinophilic and neutrophilic inflammation (Extended Data Fig. 2d). Along with the new results, we added a description to the Result section (Line 152-157).

Comment 4. *Fig 2c is unclear. Which color represents what? Is this latent time? Pseudotime? Both combined and individual plots seem to show that cluster 4 & 8 are end states, so how can they give rise to cluster 2 & 5, which also seem end states? The same can be stated for cluster 2 which is stated to be a state before cluster 5, yet they share the same latent time/pseudotime.*

Response. We apologize for unclear description. We have revised the Figure 2 legend as follows. ‘RNA velocity analysis of cDC2s at each time point and combination. Maturation stages inferred by scVelocity latent time are indicated by colors ranging from unspliced immature RNA (purple) to spliced mature RNA (yellow).’ (Line 533-535)

Comment 5. *175-186 : but why do they lose CD301b? There is no inflammation. The scdata suggests that only during OVA/HDE the CD301b should go down? Maybe there is insufficient "space" or "niche" and associated factors for these CD301b DCs to retain their phenotype?*

Response. We appreciate the reviewer’s careful reading of this section. Because the donor CD301b⁺ cDC2s were purified from C57BL/6J mice that were sensitized with OVA/HDE, it is possible that some of those cells had become activate and were transitioning to CD200⁺ DCs. However, we cannot exclude the alternative possibility that some CD301b⁺ cDC2s became activated by mechanical stress or enzymatic digestion during cell isolation. While we agree that a more detailed analysis of the mechanisms underlying cDC2 maturation would be interesting, we feel it is beyond the scope of the current study, which is centered on the lineage and developmental order of cDC2 subsets. Nonetheless, we have clarified our method in the manuscript by adding the description ‘CD301b⁺ cDC2s were prepared from OVA/HDE sensitized CD45.2 donor mice and transferred into naïve CD45.1 recipient animals (Fig. 2h, Extended Data Fig. 4b).’ (Line 198-200) as well as adding words to the figure legend ‘h, Adoptive transfer of purified CD301b⁺ lung cDC2s from OVA/HDE sensitized C57BL/6 mice (CD45.2) to naïve CD45.1 recipient mice’ (Line 538-539).

Comment 6. *Fig 3c: what does the Y-axis mean? What subsets? Is the "fold change" frequency or cell counts?*

Response. The Y-axis is the fold-change in the frequency of each depicted subset in the CD45.1⁺CD4⁺ population. The subsets are indicated by each graph title, ‘CD44^{hi}’, ‘CD25⁺’, ‘Foxp3⁺’ and ‘Foxp3⁺CD25⁺’. To clarify the Y-axis, we added a sentence to the Figure 3 legend, ‘Data shown represents fold-change of each subset frequency within the CD45.1⁺CD4⁺ or within CD45.1⁺CD44^{hi}CD4⁺ population.’ (Line 562-563)

Comment 7. *245-247 : why would these DCs be in the alveolar ducts and not around the larger airways? Data on larger airways are not provided. Maybe there are also CD301b DCs there, who sense Csf2 from ILC2s instead?*

Response. It is well-known that type 2 alveolar epithelial cells are major producers of GM-CSF in the lung, as we stated in the main text (Line 277-279) ⁶. As CD301b⁺ cDC2s are GM-CSF-dependent, it seems reasonable to us that they would be in close proximity to GM-CSF-producing AT2 cells. We have a description about localization of CD301b⁺ cDC2s and alveoli consistent with AT2 cells (Line 281-283).

We did find CD301b⁺ cells around the airway. However, majority of those cells lack CD11c and have high levels of F4/80, thus fitting the typical macrophage profile. The presence of CD301b⁺ macrophages is in agreement with gene expression profile of interstitial macrophages described above in our response to Reviewer 2, Comment 2. To address this concern, we added new panels of images displaying CD301b⁺F4/80⁺ macrophages around the airway (see Extended Data Fig. 7b). On this note, it is interesting that another GM-CSF-dependent cDC subset, cDC1s, is also closely associated with the airway. We cannot exclude that GM-CSF produced by ILCs might contribute to the development and/or survival of these DCs. To support the potential role of ILC-derived GM-CSF, we added new panels of images displaying CD103⁺CD11c⁺F4/80⁻ cDC1s around airway (Extended Data Fig. 7c), described this in the Results section (Line 283-288) and in the Methods section (Line 814). To clarify the difference between macrophages and cDCs, we added F4/80 to our histological analysis of cDC2s, and replaced the images in Figure 4h. The new images demonstrate that CD301b⁺ cells around alveolar duct are CD11c⁺ and F4/80-negative or low, suggesting these cells are cDC2s.

Comment 8. *250-252 Itgax-cre x Csf2rb : CD11c-cre targets the entire myeloid fraction, so all myeloid cells will lack Csf2rb. Alveolar macrophages are highly disturbed in this model. Maybe an inflammatory setting is created due to continuous AM turnover (death & monocyte influx) which causes the decrease in CD301b DCs? In my view, a Zbtb46-Cre would be better if available? In any case, this downside of the model should be clearly stated.*

Response. We thank the reviewer for the insightful comment. Macrophage numbers were indeed decreased in lungs of *Csf2rb*^{ADC} (*CD11c*^{Cre} x *Csf2rb*^{fx}) mice (the result is now presented in Extended Data Fig. 8a in revision), and this reduction could potentially influence cDC numbers through an indirect pathway. In response to this reviewer's suggestion, we now clarify that macrophages are also decreased in *Csf2rb*^{ADC} mice (Line 293-296). To address this issue, we followed this reviewer's suggestion and crossed *Csf2rb*^{fx/fx} mice with *Zbtb46*^{Cre} mice to generate a new mouse strain, *Csf2rb*^{AZbtb46}, which selectively lacks *Csf2rb* in cDCs. As anticipated, *Csf2rb*^{AZbtb46} mice have normal level of macrophages (not shown). However, total lung cDC2s, and specific cDC2 subsets, CD301b⁺, CD200⁺ and Ly6C⁺ cells were not clearly decreased in *Csf2rb*^{AZbtb46} mice compared with *Csf2rb*^{fx} control mice, as presented in below graphs.

To test whether the floxed *Csf2rb* gene was efficiently excised in *Csf2rb*^{AZbtb46} (*Zbtb46*^{Cre} *Csf2rb*^{fx/fx}) mice, we evaluated display of the CSF-2 receptor B (CSF2RB)/CD131 protein on the cell surface of cDC2. Surprisingly, CSF2RB protein remained on the cell surface of lung cDC2s including CD301b⁺ cDCs in *Csf2rb*^{AZbtb46} mice at comparable levels with *Csf2rb*^{fx} mice, although CSF2RB protein level was significantly reduced in *Csf2rb*^{ADC} (*Itgax*^{Cre} *Csf2rb*^{fx/fx}) mice compared with the floxed control and *Csf2rb*^{AZbtb46} mice (shown in below graphs). This inefficient excision of *Csf2rb*

gene in *Csf2r^{AZbtb46}* mice could be due to low expression of *Zbtb46* in an immature stage of cDC2, including preDCs and Ly6C⁺ cDC2s, as we previously reported ⁴. Indeed, CSF2RB protein level on preDCs was not reduced in *Csf2rb^{AZbtb46}* mice compared to *Csf2rb^{fx}* control mice (data not shown). It seems likely that this very minor reduction of CSF2RB on mature cDCs does not influence their maintenance after maturation. Thus, despite our best efforts to address this question through a different Cre recombinase system, in the end it was not helpful, and we have not included the data in the present manuscript. However, as described in our responses to Reviewer 1, Comment 1, we have addressed this point in a different way, namely by performing adoptive transfer of macrophages to restore their numbers in *Csf2rb^{ADC}* mice. We found that while this transfer restored macrophage numbers in these animals, it did not restore CD301b⁺ DC numbers and Treg inducing activity, thus providing additional support for an important role for CD301b⁺ DCs in this regard. These new results are presented in Extended Data Figure 8b, 8c and 8d, and a description was added to the main text (Line 304-311 and 322-325).

Comment 9. 265-274 : *To account for the issue mentioned above this experiment (culturing of DCs lacking Csf2rb) should be done with BM DCs maybe or DCs from skin? Right now the ENTIRE cDC2 fraction is used, but this fraction has already been influenced by the disturbed environment due to AM turnover.*

Response. We thank the reviewer for their insightful suggestion. To evaluate DC function in the absence of a potential bystander effect of lung macrophages, we generated BMDCs from WT mice, then separately purified CD301b⁺ and CD200⁺ BMDC2s, and cultured them with naïve CD4⁺ T cells from *Foxp3^{GFP}* reporter mice. The efficient Treg induction by CD301b⁺ BMDC2s that was presented in our original submission is now presented in Figure 7c.

To test whether CD301b⁺ BMDC2s can induce tolerance *in vivo*, we adoptively transferred CD301b⁺ BMDC2s to mice after allergic sensitization, but prior to challenge. OVA-bearing CD301b⁺ BMDC2s significantly suppressed eosinophilic inflammation of the airway (Figure 7d and e). This result supports that the tolerogenic function of GM-CSF-dependent cDC2s does not depend on macrophages. Along with the new results, we added a description to the Results section (Line 404-411).

To confirm the requirement of GM-CSF for the development of CD301b⁺ BMDC2s, we generated cDCs from *Csf2rb^{fx}* and *Csf2rb^{ADC}* mouse bone marrow. Consistent with lung cDC2s, CD301b⁺ BMDC2s were fewer in *Csf2rb^{ADC}* cells compared with *Csf2rb^{fx}* cells (Extended Data Fig. 11b) probably because GM-CSF only moderately stimulated DC precursors through its low affinity receptor CSF2RB2. Descriptions of those data have now been added to the Results section (Line 384-392 and 398-401).

Reviewer #3

Manuscript number: NCOMMS-24-30321-T

Title: "GM-CSF-dependent CD301b+ lung dendritic cells confer 1 tolerance to inhaled allergens" by Christina L. Wilkinson et al. In the present manuscript by Christina L. Wilkinson et al., the authors intended to identify dendritic cell (DC) subpopulations that are responsible for CD4+FOXP3+ regulatory T (Treg) cell induction in mouse lungs. Using OVA-sensitization and preclinical mouse models of allergic asthma, the authors identify CD301b+ DCs that are comparatively strong in inducing Treg cell development or expansion. Employing conditional knock out mouse models the authors demonstrate that development/Expansion of CD301b+ DCs depends on GM-CSF. Further, the authors show that under inflammatory conditions CD301b+ DCs can differentiate into CD200+ DCs promoting Type 2 immunity and TH2 differentiation and that transfer of CD301b+ T cells might offer potential for immunological tolerance induction to inhaled allergens.

Validity

Data presented in this paper are thoroughly analyzed and statistical testing is performed where indicated. Data are not over-interpreted, although a few logical inconsistencies have been detected and improvements are suggested below. In general, there is no concern about the validity of the findings. This is a well-written manuscript providing strong evidence for a role of CD301b+ DCs in Treg cell induction in the lungs of mice. However, several concerns rose during the review:

Analytical approach

No concerns.

Response. Thank you for careful reading and insightful comments. Along with reviewer's suggestions, we performed experiments and analyses. Some new results and descriptions were added to the manuscript. Some results are shown in this response. Please see our point-by-point responses shown below. The revised descriptions along with additional results are highlighted by yellow color in the main text, and relevant references are listed at the end of this response document.

Major critique:

Comment 1. *Based on the expression of HELIOS, ROR γ t, GATA-3 and other transcription factors, CD4+FOXP3+ regulatory T cells can be subdivided and better characterized. Since the authors emphasize in this manuscript that CD301b+ DCs influence the development of peripheral (p)Treg cells, the authors should be motivated to better characterize CD301b+ DC-induced Treg cells at the level of transcription factor expression. In addition, the authors should measure ability of the different DC subsets described in this manuscript for their capacity to induce proliferation of different Treg cell subsets.*

Response. Thank you for these helpful comments. In accordance with this reviewer's suggestion, we used flow cytometry to measure HELIOS, GATA3 and ROR γ t in CD4⁺ T cells after coculture with purified CD301b⁺ cDC2s. We found that the majority of Foxp3⁺ T cells express HELIOS, a lesser fraction expresses GATA3, and a small number of cells express ROR γ t. These results are presented in Extended Data Fig. 2b. As the expression of HELIOS stabilizes the suppressive function of Tregs, co-expression of Foxp3 and HELIOS suggests that Tregs induced by CD301b⁺ cDC2s are likely immunosuppressive. Along with the new results, descriptions were added to the Results section (Line 128-136) and the Methods section (Line 739-746).

Comment 2. *Next to demarcating peripheral Treg cells from Treg cells of thymic origin, subsets have been described based on their capacity to suppress and/or to promote tissue repair. Given the role of tissue repair Treg cells in type 2 immunity, the authors should be encouraged to functionally phenotype the Treg cells induced by CD301b+ DCs.*

Response. Thank you for the suggestion, as testing the function of CD301b⁺-induced Treg cells would be interesting. As described in the above response, Tregs induced by CD301b⁺ cDC2s co-express Foxp3 and HELIOS (Extended Data Fig. 2b). Although HELIOS expression in periphery-induced Tregs (pTreg) is still controversial, important roles of this transcription factor for regulatory function of Tregs has been clearly demonstrated and widely accepted. We believe that co-expression of Foxp3 and HELIOS but not effector T cell-transcription factors is supportive evidence to interpret that CD301b⁺ cDC2-induced Foxp3⁺ CD4⁺ T cells are bona fide regulatory T cells. A new sentence was added to the main text to reflect this (Line 135-136).

Comment 3. *In figures 1c and d, 3c, 5e, 6c and 7c the authors test the ability of different DC subsets from wt and transgenic mice to induce FOXP3 expression in CD4⁺ T cells. What's the fate of the FOXP3-negative CD4⁺ T cells under these various conditions in terms of other T cell lineage determining transcription factors like GATA-3, T-bet, RORgt? In the same vein and as previously asked, it would be interesting to know which transcription factors the FOXP3⁺ Treg cells still express (HELIOS, GATA-3, RORgt).*

Response. Thank you for the interesting insight. Along with reviewer's suggestion, we analyzed the expression of HELIOS, GATA3, and RORγt in FoxP3⁻ CD4⁺ T cells after *ex vivo* coculture with CD301b⁺ lung cDC2s. A substantial fraction of these FoxP3⁻ CD4⁺ T cells expressed HELIOS, a small fraction expressed GATA3, and minimal cells expressed RORγt. These results were similar with the FoxP3⁺ CD4⁺ T cells generated in *ex vivo* culture with CD301b⁺ cDC2s. This might be due to sensitivity issue in detection of intracellular Foxp3, or HELIOS expression in some effector T cells⁷. The differentiation of effector T cells is scientifically interesting, but it raises further questions, and we feel that it is beyond the scope in the present study, as our main focus is in Treg induction. Therefore, we decided to do not present the results of Foxp3⁻ CD4⁺ T cells in this manuscript.

Comment 4. *In the same sense, in figure 1f the authors analyzed total cell count, eosinophils, neutrophils and lymphocyte upon transfer of CD301b⁺ and CD301b⁻ DC prior to OVA/HDE sensitization. How about expression of FOXP3, GATA-3, T-bet and RORgt in CD4⁺ T cells under these conditions?*

Response. Thank you for the comment on CD4⁺ T cell subsets. We analyzed transcription factors in lung CD4⁺ T cell 5 days after immunological tolerance induction by giving adjuvant-free ovalbumin. Although this experimental design is slightly different from reviewer's suggestion, we believe this is a good model to measure T cell subsets under tolerogenic conditions. As shown in the graph below, Foxp3⁺ and HELIOS⁺ cells are dominant among CD44^{hi} activated

CD4⁺ T cells in the lung. While the induction of CD4⁺ T cell subsets under pathogenic conditions is an interesting topic, we feel it is beyond the scope of the present study. Therefore, we did not pursue the analysis further.

Comment 5. *In previous reports it was shown that Klf4 (Tussiwand et al. Immunity 2015) or PD-1 and IRF-4 expressing DC are inducers of type 2 immunity (Williams et al. Nat Comm 2013, Gao et al. Immunity 2013). How can CD200+ DCs be differentiated from this Klf4 or IRF-4 and PDL-1 expressing DC subset and what is the expression of these molecules in CD301b+ DC?*

Response. Thank you for the insightful comment. As this reviewer mentioned, previous papers demonstrated important roles of IRF4, KLF4 and PD-1 in the induction of type 2 immunity. Of note, these previous studies tested gene expression and function of total cDC2s. To investigate subset-specific gene expression of *Klf4*, *Irf4*, *Cd274* (encoding PD-L1), and *Pdcd1* (encoding PD-1), we analyzed our scRNA-seq data (shown below). We found that *Klf4* was expressed mostly in clusters 4, 8 (*Mgl2*-expressing clusters) as well as clusters 7 and 1 (*Ly6c*-expressing clusters), *Irf4* was expressed among all cDC2 subsets, while *Cd274* is mostly in clusters 2 and 5 (*Cd200*-expressing clusters). *Pdcd1* expression was minimal among all cDC clusters probably because this gene is expressed by T cells. These results are consistent with our previously published results. Some transcription factors could be important for development of Th2-inducing cDC2s at some points during their maturation, but their expression is not specifically limited to Th2-inducing cDC2 subset.

To test the role of IRF4 for the development of lung cDC2 subsets, we analyzed lung cDCs in mice lacking *Irf4* in CD11c-expressing cells (*Irf4*^{ΔDC}), which were generated by crossing *Itgax*^{Cre} and *Irf4*^{fl/fl} mice. As shown in the graphs below, both at steady state and after allergic sensitization with OVA/HDE, lung cDC1 numbers were similar in IFR4-sufficient (*Irf4*^{fl/fl}) mice and *Irf4*^{ΔDC} mice, whereas cDC2s were decreased in *Irf4*^{ΔDC} mice. Among the cDC2 subsets, we found that all subsets were decreased in *Irf4*^{ΔDC} mouse lungs, indicating that this transcription factor is important for development of all cDC2 subsets. These results are congruent with our findings shown in Figure 2 suggesting that these subsets are developmentally related. While CD200⁺ cDC2s promote Th2 differentiation, other cDC2 subsets can stimulate other T helper cell lineages under different conditions. These results are interesting, but further study of the mechanisms underlying effector T cell induction is beyond the scope of the present study. Therefore, we decided to not present these results in our revised manuscript.

Comment 6. In figure 6 the authors nicely demonstrate the role of CD301b⁺ DC-produced TGF-beta in the induction of Treg cells. Since Treg cell induction relies on TGF-b and IL-2 it would be interesting to understand the source of IL-2 in this context. In the same vein, next to Treg cell differentiation TH17 cell differentiation can also be promoted by TGF-b in context with IL-6. Do CD301b⁺ DCs also promote TH17 cell differentiation?

Response. Thank you for this insightful comment. To examine the source of IL-2, we used flow cytometry to assess intracellular IL-2 in CD4⁺ T cells and cDC2 subsets 2 days after their co-culture. As shown in below (left), CD301b⁺ cDC2s promoted the highest percentages of IL-2 positive T cells. By contrast, relatively few DC2s contained intracellular IL-2 (right). This suggests that CD4⁺ T cells are likely the primary source of IL-2. The IL-2 induction by

CD301b⁺ cDC2-stimulated T cells is consistent with our findings that this cDC subset preferentially stimulates Treg differentiation.

We previously studied Th17-induction by lung cDC subsets, and reported that Ly6C⁺ cDC2s preferentially promote Th17 differentiation by producing IL-1 β and IL-6⁴, while CD301b⁺ cDC2s are inefficient in this regard. We have described the previous finding in the Introduction (Line 69-71).

Comment 7. *In figure 7 the authors demonstrate that adoptive transfer of CD301b⁺ DCs can confer immunological tolerance. In their introduction to authors refer to SCIT and SLIT as treatment possibilities to induce allergen-specific tolerance. Does SCIT and/or SLIT induce expansion/ de novo generation of CD301b⁺ DC via GM-CSF in humans?*

Response. Induction of CD301b⁺ cDC2s via GM-CSF by SCIT and/or SLIT treatment in humans is an interesting idea. There are no studies measuring specific DCs or cytokine production post-treatment, but there have been several scRNA-seq studies demonstrating the suppression of pathogenic Th2 cell responses. Specifically, a study collected PBMC samples from patients with cedar pollen allergy before and after SLIT treatment. The group found that Th2 cells from patients after SLIT had increased musclin (*MSC*), *TGFB1*, and *IL2*, which are known markers for Treg differentiation. These data suggest that that expansion of Th2 cells after treatment resulted in trans-type Th2 cells suppressing allergic response⁸. We added a sentence ‘Immunotherapies in the form of SCIT and SLIT can be very effective; scRNA-seq of PMBCs from patients before and after SLIT treatment leads to clonal expansion of allergen-specific Tregs as well as trans-type Th2 cells that express high *musculin*, *TGF- β* , and *IL-2*⁸. ’ to the Discussion (Line 418-421).

SCIT and/or SLIT treatment might induce GM-CSF in the skin. As we discussed in the main text (Line 482-488), a previous report demonstrated that production of GM-CSF in peripheral tissues enhances allergic inflammation through cDC activation^{9 10, 11}. GM-CSF is required for development of tolerogenic cDCs, but this cytokine can activate mature cDCs that already present in peripheral tissues. Furthermore, since skin CD301b⁺ cDC2 are known to promote Th2 differentiation, mechanisms of tolerance induction could be different between skin and airway. To avoid overstatement without evidence, we decided to do not describe potential induction of CD301b⁺ cDC2s by SCIT or SLIT treatment via GM-CSF production in the present manuscript.

Minor critique:

Comment 8. *Given the role of CD301b⁺ DC in Treg cell induction I was wondering whether Csf2deltaARE mice harbor higher Treg cell numbers already under steady state conditions in the lungs?*

Response. Thank you for insightful comment. We measured FoxP3⁺ cells among CD4⁺ T cells in the lung at steady state conditions of both *Csf2^{flx}* and *Csf2^{ARE}* mice and found that FoxP3⁺ Tregs are significantly more abundant in GM-CSF overexpressing mice. The results were presented in Figure 4g, and we added a description to the Results section (Line 271-276).

References

1. Suzuki, T. *et al.* Pulmonary macrophage transplantation therapy. *Nature* **514**, 450-454 (2014).
2. Bourdely, P. *et al.* Transcriptional and Functional Analysis of CD1c(+) Human Dendritic Cells Identifies a CD163(+) Subset Priming CD8(+)CD103(+) T Cells. *Immunity* **53**, 335-352.e338 (2020).
3. Villani, A.C. *et al.* Single-cell RNA-seq reveals new types of human blood dendritic cells, monocytes, and progenitors. *Science* **356** (2017).
4. Izumi, G. *et al.* CD11b(+) lung dendritic cells at different stages of maturation induce Th17 or Th2 differentiation. *Nat Commun* **12**, 5029 (2021).
5. Tsuiji, M. *et al.* Molecular cloning and characterization of a novel mouse macrophage C-type lectin, mMGL2, which has a distinct carbohydrate specificity from mMGL1. *J Biol Chem* **277**, 28892-28901 (2002).
6. Gschwend, J. *et al.* Alveolar macrophages rely on GM-CSF from alveolar epithelial type 2 cells before and after birth. *J Exp Med* **218** (2021).
7. Akimova, T., Beier, U.H., Wang, L., Levine, M.H. & Hancock, W.W. Helios expression is a marker of T cell activation and proliferation. *PLoS One* **6**, e24226 (2011).
8. Iinuma, T. *et al.* Single-cell immunoprofiling after immunotherapy for allergic rhinitis reveals functional suppression of pathogenic T(H)2 cells and clonal conversion. *J Allergy Clin Immunol* **150**, 850-860.e855 (2022).
9. Nobs, S.P. *et al.* GM-CSF instigates a dendritic cell-T-cell inflammatory circuit that drives chronic asthma development. *J Allergy Clin Immunol* **147**, 2118-2133 e2113 (2021).
10. Cates, E.C. *et al.* Effect of GM-CSF on immune, inflammatory, and clinical responses to ragweed in a novel mouse model of mucosal sensitization. *Journal of Allergy and Clinical Immunology* **111**, 1076-1086 (2003).
11. Stämpfli, M.R. *et al.* GM-CSF transgene expression in the airway allows aerosolized ovalbumin to induce allergic sensitization in mice. *The Journal of Clinical Investigation* **102**, 1704-1714 (1998).